# Recent Advancements in Gallic Acid-Based Drug Delivery: Applications, Clinical Trials, and Future Directions

**DOI:** 10.3390/pharmaceutics16091202

**Published:** 2024-09-13

**Authors:** Ranjit K. Harwansh, Rohitas Deshmukh, Vijay Pratap Shukla, Dignesh Khunt, Bhupendra Gopalbhai Prajapati, Summya Rashid, Nemat Ali, Gehan M. Elossaily, Vijendra Kumar Suryawanshi, Arun Kumar

**Affiliations:** 1Institute of Pharmaceutical Research, GLA University, Mathura 281406, India; harwanshranjeet@gmail.com (R.K.H.); rahi18rahi@gmail.com (R.D.); vijaypratapshukla28@gmail.com (V.P.S.); 2School of Pharmacy, Gujarat Technological University, Gandhinagar 382027, India; digneshkhunt80@gmail.com; 3Shree S. K. Patel College of Pharmaceutical Education and Research, Ganpat University, Mehsana 384012, India; bhupen27@gmail.com; 4Faculty of Pharmacy, Silpakorn University, Nakhon Pathom 73000, Thailand; 5Department of Pharmacology & Toxicology, College of Pharmacy, Prince Sattam Bin Abdulaziz University, P.O. Box 173, Al-Kharj 11942, Saudi Arabia; s.abdulrashid@psau.edu.sa; 6Department of Pharmacology and Toxicology, College of Pharmacy, King Saud University, P.O. Box 2457, Riyadh 11451, Saudi Arabia; nali1@ksu.edu.sa; 7Department of Basic Medical Sciences, College of Medicine, AlMaarefa University, P.O. Box 71666, Riyadh 11597, Saudi Arabia; jabdelmenam@um.edu.sa; 8Faculty of Pharmacy, Bharti Vishwavidyalaya, Durg 491001, India; khanna.vijendra@gmail.com; 9School of Pharmacy, Sharda University, Greater Noida 201310, India

**Keywords:** gallic acid, nanotechnology, bioavailability, nanocarrier system, permeability

## Abstract

Gallic acid (GA) is a well-known herbal bioactive compound found in many herbs and foods like tea, wine, cashew nuts, hazelnuts, walnuts, plums, grapes, mangoes, blackberries, blueberries, and strawberries. GA has been reported for several pharmacological activities, such as antioxidant, inflammatory, antineoplastic, antimicrobial, etc. Apart from its incredible therapeutic benefits, it has been associated with low permeability and bioavailability issues, limiting their efficacy. GA belongs to BCS (Biopharmaceutics classification system) class III (high solubility and low probability). In this context, novel drug delivery approaches played a vital role in resolving these GA issues. Nanocarrier systems help improve drug moiety’s physical and chemical stability by encapsulating them into a lipidic or polymeric matrix or core system. In this regard, researchers have developed a wide range of nanocarrier systems for GA, including liposomes, transfersomes, niosomes, dendrimers, phytosomes, micelles, nanoemulsions, metallic nanoparticles, solid lipid nanoparticles (SLNs), nanoparticles, nanostructured lipid carriers, polymer conjugates, etc. In the present review, different search engines like Scopus, PubMed, ScienceDirect, and Google Scholar have been referred to for acquiring recent information on the theme of the work. Therefore, this review paper aims to emphasize several novel drug delivery systems, patents, and clinical updates of GA.

## 1. Introduction

Gallic acid (GA) is a polyphenolic herbal molecule that has been reported for several health benefits. GA and its derivatives are present in medicinal plants and have been famous for their several pharmacological activities [1]. Gallic acid is a yellowish-white, crystalline phenolic molecule that has been demonstrated to exhibit a wide range of important biological activities, primarily because of the phenolic hydroxyl group. The alkaline readily produces GA, or acidic hydrolysis of tannin, which is present in large quantities in plants. GA derivatives include catechin and ester derivatives [2]. Alkyl esters, which are mostly made up of methyl gallate (MG), propyl gallate (PG), octyl gallate (OG), dodecyl gallate (DG), tetradecyl gallate (TG), and hexadecyl gallate (HG), are the most common ester derivatives of GA. Epicatechin (EC), epicatechin gallate (ECG), gallocatechin gallate (GCG), and epigallocatechin (EGC) are some of the most widely used derivatives of catechins [3,4,5]. Important biological activities of GA include its anti-inflammatory, antimicrobial, antifungal, anticarcinogenic, antityrosinase, antitubercular, antidiabetic, cardioprotective, etc. Its ability to eliminate free radicals efficiently makes it a potent antioxidant [6]. Various functions and biological activities of GA are represented in Figure 1.

GA is considered a safe molecule at the oral dose of 5000 mg/kg for exhibiting therapeutic effects [6]. According to Zanwar et al. and Ngamukote et al., GA suppresses pancreatic cholesterol-esterase activity by attaching to bile acids and decreasing the dissolution of cholesterol in micelles [7,8]. Therapeutic effects of GA have been described for various disorders, including the digestive system, nervous system, metabolism, and cardiovascular system [9].

GA is classified as a BCS (Biopharmaceutics classification system) class III substance, meaning it has low drug permeability and is very soluble, suggesting a permeability problem. A little research has been done on gallic acid’s bioavailability in humans. Gallic acid appears to be an incredibly well-absorbed phenolic compound compared with other polyphenols. The most frequently found metabolic product of GA within human biological fluids, 4-O-methyl gallic acid, has been shown to rise rapidly in human plasma over the first four hours after only one dose of red wine ingestion. The amounts of 4-O-methyl gallic acid and the unbound and glucuronidated variants of GA within the plasma elevated when 50 mg of unadulterated GA was ingested [10].

The bioavailability of metoprolol was improved when co-administered with GA and ellagic acid orally. In an animal model, enhanced bioavailability and therapeutic effect of metoprolol were achieved through inhibition of Cytochrome P450 family two subfamily D member 6 (CYP2D6)-mediated metabolism. The outcome of an independent investigation was that we assessed the oral bioavailability of the drug in male Wistar rat models. As a result, adverse herbal–drug interactions may occur if ellagic acid and gallic acid supplements are taken concurrently with drugs that are CYP2D6 substrates [11]. Hence, precautions should be taken before administering GA and other medications.

GA is metabolized by different biotransformation processes, resulting in quick elimination. GA has a short half-life, low therapeutic efficacy, and poor bioavailability, which limits its use in clinical practice [10]. In this context, a novel drug delivery system (NDDS) is pivotal in solving solubility, permeability, and therapeutic efficacy [12]. In the last few decades, several nanocarriers of GA have been developed, like liposomes, transfersomes, phytosomes, nanoemulsions, micelles, nanoparticles, green synthesized nanoparticles, etc. [13]. NDDSs can load the drug inside the carriers and channel their release to a target site to get the desired result.

Moreover, NDDSs can improve drugs’ physical and chemical stability by encapsulating them into suitable nanocarriers. They can also prevent the drug loss from gastric degradation [14]. Thus, the review article addresses the challenges, pharmacological role, various NDDSs, patents, and clinical trial status of GA.

## 2. Impact of Gallic Acid on Microbiome and Oxidative Stress

### 2.1. Microbiome

The human microbiota is made up of approximately 10–100 trillion symbiotic microbial cells—primarily gut bacteria—that circulate by each individual [15]. Gut bacteria can prevent many illnesses, including swelling, malignancies, ulcers, cancer, obesity, diabetes, hepatic problems, etc. Several health issues are becoming more prevalent in today’s lifestyle due to poor diet and underutilization of nutraceuticals [16].

The human microbiome is made up of the microbial, viral, and eukaryotic microbes that reside in the body. These microbes can affect human physiology in both healthy and diseased states. They affect most of our physiological functions, directly or indirectly, through their assistance of metabolic processes, protection against pathogens, and immune system education [17].

According to recent research, GA and its metabolites improve the functioning of the microbiome in the gut and impact immune responses. GA consequently possesses a great deal of the ability to modify the immune system and strengthen the defenses within the body against microbial illnesses [4]. By increasing antioxidant enzymes like GPx (glutathione peroxidase) and superoxide dismutase (SOD), GA has been shown to lower myocardial stress. Moreover, GA has been shown to reduce tumor necrosis factor and proliferation, hence reducing inflammation associated with atherosclerosis [18].

In an experiment, the feces microbiota and metabolomics of 45 days of an essential diet enriched with GA (0%, 0.02 percent, 0.04%, and 0.08%)-fed healthy dogs were evaluated to establish the potential risk for a long-time GA intake to gut health. This investigation showed that the GA supplementation modulated blood lipid metabolism by reducing serum triglycerides, fat digestion, and the overall bacteroidetes/firmicutes ratio [19]. In a recent study, it has been discovered that the gut microbiota might affect the capacity of mutant p53 to cause the development of intestinal tumors [20].

In a study, rats that voluntarily ingested 20% ethanol on different days for 13 weeks displayed higher Bacteroidetes, lower Lactobacilli abundance, and reduced α- and β-diversity compared with the non-exposed control group. In a previous study, mice given a lower dose of ethanol (0.8 g/kg/day) for seven days demonstrated decreased α-diversity in their feces microbiome and phylum bacterial species frequency compared with the control group. Bacteroidetes increased while Proteobacteria decreased in the intestines of rhesus macaques treated with water containing 4% ethanol [21].

### 2.2. Oxidative Stress

Oxidative stress can result from an imbalance between the body’s ability to remove and purify reactive oxygen species (ROS) and the quantity of ROS generated. Reactive oxygen intermediates include anion superoxide (O_2_^•−^), peroxide of hydrogen (H_2_O_2_), and radicals of hydroxyl (HO^•^) [22]. These chemicals can damage nucleic acids, proteins, and membranes. Oxidative stress arises from these intermediates. There is growing evidence that the damage caused by reactive oxygen species builds up over time and has a role in numerous diseases [23]. For various reasons, the pro-oxidant–antioxidant equilibrium concept is central to our understanding of oxidative stress. First, it draws attention to the potential that changes on either end of the equilibria (such as an excess of reactive oxygen compound generation or a deficiency in antioxidant defenses) could cause the disturbance. It also highlights how many illnesses are associated with homeostatic ROS species concentrations [24]. In the fight against oxidative stress-related diseases like glaucoma, diabetes, tumors, memory loss, inflammation, renal and liver failure, etc., herbal biological agents and other natural remedies are making revolutionary strides [17].

GA’s capacity to control oxidative stress and influence the oxidoreductive status of tumor cells has been linked to its anticancer properties. It can autoxidize in malignant cells to produce significant quantities of H_2_O_2_ and O_2_. These increased ROS levels may cause the activation of caspases 3, 8, and 9, the release of cytochrome c, and the loss of mitochondrial potential. Consequently, gallic acid treatment could effectively cause cancer cells to undergo apoptosis [25]. GA may halt the proliferation of the human colon cancer cell line (HCT-15) by regulating ROS-dependent cellular death. Furthermore, it is widely recognized that a drop in glutathione (GSH) and an increase in ROS are associated with the mortality that GA causes in A549 lung cancer cells. Additionally, proapoptotic (Bax and Bad) as well as antiapoptotic (B-cell lymphoma-extra-large; Bcl-x and Bcl-2) protein-coding genes may be impacted by GA [26]. 

An investigation conducted on rats revealed that the rats were made drunk by giving 600 parts per million of sodium fluoride (NaF) in their water for a week. A week before NaF exposure, the positive control, silymarin (10 mg/kg), and galactose (10 and 20 mg/kg), were administered. Lipid peroxidation, decreased levels of glutathione, and the activity of liver SOD and catalase were assessed 24 h post-treatment. Serum biochemical indicators, including aspartate aminotransferase, lipase, glucose, albumin, triglycerides, total bilirubin, and direct bilirubin, likewise were measured. The preliminary treatment with gallic acid was shown to normalize the levels of oxidative stress in tissues of the liver and the alterations in blood parameters caused by sodium fluoride [27].

In the event of elastase-induced lung damage, the author postulated that gallic acid, due to its anti-inflammatory and antioxidant properties, would shield the lung and the associated cardiac dysfunction. One hundred eight male rats (Sprague–Dawley rats) were divided into six groups at random: three groups with different doses of gallic acid significantly (GA 7.5, GA 15, and GA 30 mg/kg), control, porcine pancreatic enzyme (PPE), and PPE+GA. On days 1 and 10 of the test, PPE was administered intratracheally. Exams included each group’s oxidative stress, bronchoalveolar lavage fluid, hemodynamic measures, and electrocardiography. Following PPE delivery, hemodynamic parameters (*p* < 0.05, *p* < 0.01, and *p* < 0.001), SOD (*p* < 0.05), heart rate (HR), and QRS voltage of the electrocardiogram (ECG) all decreased. Interleukin 6, or IL-6 (*p* < 0.001), malondialdehyde (MDA) (*p* < 0.001), tumor necrosis factor-α (TNF-α) (*p* < 0.001), and the overall amount of white blood cells (*p* < 0.001) all showed a substantial increase in PPE groups. Compared with the PPE group, gallic acid maintained hemodynamic characteristics, oxidative stress, inflammation, and ECG markers. This study discovered that gallic acid, possibly due to its antioxidant and anti-inflammatory properties (Figure 2), had a strong therapeutic effect on heart dysfunction brought on by elastase-induced lung injury, such as emphysema [28].

## 3. Nanocarrier Preparation and Nanocarrier Systems of GA

The creation of nanoparticles (NPs) is essentially the result of all of these procedures, which may be generally divided into top-down and bottom-up tactics. The conditions surrounding the synthesis define the fundamental properties of NPs. Size and size distribution, directional qualities, crystallinity, mutual alignment, and shape are a few of these attributes. The following section provides coverage of the two fundamental kinds of NP synthesis methods.

### 3.1. Bottom-Up Strategies

One of the most popular methods for producing NPs is chemical reduction. This method also makes it possible to adjust the reactant’s feed rate, dispersion, and molar concentration, producing nanoparticles with regulated forms, sizes, and distributions of sizes [29]. Another intriguing technology is the quick and inexpensive way to produce NPs electrochemically on a substrate. Adjusting the electrochemical conditions can produce NPs of different sizes and forms. This method can be utilized with a wide variety of ions. Layer by layer, atoms must develop in bottom-up electrochemical methods [30]. Electrochemical reduction is one method used to create hybrid NPs, such as graphene–AuNPs. Graphene sheets are subjected to an electrochemical potential after being placed on an electrode and immersed in a solution of electrolytes comprising metallic precursors. This is the way that graphene–AuNPs are often made. The photochemical method, which produces NPs utilizing photochemical sources for potent reducing radicals, is another way to synthesize NP. It enables spatiotemporal modulation of NP formation by controlling the size of particles with light intensity. The extraordinary stability these particles display can be achieved without stabilizing chemicals [31]. 

NPs can be produced using high-intensity ultrasound without requiring extreme temperatures, elevated pressures, or long reaction times [3]. Several theories have been put up to explain how 20 kHz sonic irradiation breaks molecular bonds. The critical process in sonochemistry is cavitation; in other words, bubbles’ generation, growth, and collapse are related. The cavitational collapse produces incredible heating/cooling speeds, enormous local temperature, and pressure, which provide a unique mechanism for manufacturing high-energy chemistry [32]. The hydrothermal method is another kind of synthesis that relies on synthesis in the solution phase. As stated differently, it’s a process of converting room-temperature solutions into hot ones. According to the vapor pressure of the substance in the solution, high and low pressure can be used to alter the morphology of the resulting nanoparticles. Using this method, components that are naturally unstable at ambient temperature can be transformed into NPs [33]. 

Microwave-aided synthesis is a popular method of creating NPs. Fundamentally, microwaves appear to replace conventional energy sources and heating methods [34]. Atomic layer deposition (ALD) has emerged as a powerful tool for synthesizing and functioning nanomaterials with unmatched benefits when contrasted with other thin film coating techniques, such as physical and chemical vapor deposition (CVD and PVD). ALD can precisely deposit materials on a solid surface by a sequence of autonomous, self-terminating chemical processes involving precursor vapors. On practically any complicated substrate, including ones with high aspect ratio nanopores higher than 1000:1, ALD’s surface regulation capability enables the creation of impressively homogenous and conformal films. It presents excellent opportunities for various applications, such as medical treatment, detectors, and magnetic recording in medicine [35]. 

Employing atomic-level regulation of self-limited chemical reactions among the substrate’s solid and gaseous states. The technique known as ALD is used to apply skinny coatings to flat substrates. The applications for the generated NPs are numerous [36]. The sol–gel procedure is the other widely recognized way to produce NPs. It depends on the metal oxide solution or metal alkoxide condensing and hydrolyzing. One kind of colloidal fluid produced by hydrolysis is called sol; another type produced when sol is condensed into a semi-solid phase is called scenario gel. Consequently, to make the necessary NPs [37,38], high heat is required to dry the gel. Inorganic nanoparticles can be created on the appropriate substrate surface by chemical vapor deposition, or CVD. Many intermediary products happen during the process of adhering to the substrate surface. The last step of the procedure breaks down the intermediate products to yield solid grains and NP [39]. 

### 3.2. Top-Up Strategies

To make NPs, this strategy or process requires that the bulk material be cleared or separated. A summary of some well-known top-down synthesis methods is provided below. One way to formulate NPs is by ball milling. This method makes alloys by spreading metals into other elements. It causes a powerful collision and impact between the balls and the powder. Typically, this procedure is used because it is inexpensive and can be performed on nearly any type of material [39,40]. 

Electrical wire explosion is another dangerous method of producing nanopowder and NPs. The process takes about 25 kWh of energy per kg and produces powders exhibiting an average particle dimension of 20–100 nm. It is also highly productive, yielding up to 200 g/h. Since the heated material expands more slowly and has more energy than binding, a higher-density current pulse passing by a piece of wire possesses greater energy density than binding energy. As a result, there’s a burst of bright light, and the wire bursts into a spray of boiling material, forming boiling droplets that shoot into the air [41]. Though highly successful, this technology uses excessive amounts of energy. For more details, go to a thorough summary of the subject that enumerates the benefits and drawbacks of different top-down and bottom-up strategies [42]. 

### 3.3. Different Nanocarrier Systems of GA

New developments in nanotechnology have sparked the development of several creative methods for delivering GA, which can be broadly categorized into the following areas: SLNs, liposomes, transferosomes, niosomes, dendrimers, ethosomes, nanocapsules, micelles, and nanogels [16]. These nanoparticulate vehicle systems serve various purposes, one of which is to boost the curative properties of GA. They do this by being non-toxic and biodegradable solid lipids, polymers, gold, and albumin nanoparticles. Targeted distribution and guided/controlled drug release are further features. The thinness and high biocompatibility of polymeric material-based nanoparticles have the advantage of being able to travel in the bloodstream for a more extended period [43]. Figure 3 shows different nanocarrier systems of gallic acid.

#### 3.3.1. Lipidic and Polymeric Nanoparticulate Carriers of GA

Lipid-based delivery systems are used for medicine safety, selective distribution, and controlled release. They are constructed from biocompatible and biodegradable lipids. The first commercial medication that utilized a lipid-based delivery system was the antimycotic drug econazole, which was released into the market in 1988. Since then, several studies demonstrating the efficacy of these distribution strategies have been published [44]. Lipid-based nanoparticles fall into two categories: SLNs and NLCs. The lipid matrix is less organized than SLNs and can hold higher drug concentrations since it combines solid and liquid lipids that melt at temperatures higher than 40 °C. This results in more flexible drug release management and prevents drug leakage during storage. The two types of nanoparticles that are beneficial for targeted distribution, pH management, and protection against environmental stressors such as pH swings, rapid fluctuations in temperature, and enzymes are nanostructured lipid carriers (NLCs) and SLNs. NLCs and SLNs are biocompatible, break down over time, and can be customized for various biological goals. Lipid nanoparticles have applications in many medical domains. They are more biocompatible than inorganic and polymeric nanocarriers [16]. 

The solid particles with colloidal structure are polymeric nanoparticles with diameters that differ by 10 nm to 1 nm. Usually, they are made of biodegradable and suitable polymers. To function as drug carriers, they encase or trap pharmaceuticals. Substances can adhere to surfaces either chemically or physically. They are effective transporters due to their outstanding stability, lengthy shelf life, water solubility, small size, and lack of toxicity. Nanospheres exhibit a matrix structure with a homogeneous drug dispersion. In nanocapsules, the medication is contained within a polymeric membrane-enclosed cavity, or it is contained inside NPs that have a membrane of polymeric material around them. The building blocks for polymeric nanoparticles are hydrophilic natural polymers like proteins and polysaccharides. To prepare polymeric nanoparticles, two kinds of polymers are used: synthetic hydrophobic polymers, which are classified into pre-polymerized polymer materials such as polystyrene and poly(ε-caprolactone), and polymerized-in-process polymer compounds like poly(methyl methacrylate) and poly(isobutyl cyanoacrylate). Proteins like albumin and gelatin, as well as polysaccharides like chitosan and alginate, are examples of naturally occurring hydrophilic polymers [45]. In a comparative study, pure gallic acid has shown various values for their pharmacokinetic parameters, like C_max_: 2.5 ± 0.18 µg/mL, AUC_0–t_: 2.82 ± 0.2 µg·h/mL, AUC_0–∞_: 3.09 ± 0.21 µg·h/mL, T_max_: 1.75 ± 0.78 h, and polymeric nanoparticles have the following values for their pharmacokinetic parameters: C_max_: 0.6957 ± 1.52 µg/mL, AUC_0–t_: 5.0617 ± 2.01 µg/mL^−1^ h, T_max_: 6 h, Ke: 0.15 h, which shows that polymeric nanoparticles have better pharmacokinetic properties than pure gallic acid and it can easily enhance the bioavailability and permeability of GA [46]. 

Producing polymeric-based nanoparticulate carriers (NPC) can be synthetic polymers, including polyamines achieved through the synthesis of synthetic or semi-polyester-based materials, poly(amino acid), poly(alkyl cyanoacrylates), polyorthoesters (POE), and polyanhydrides. A binding molecule placed among the bioactive and the NPC surface allows the bioactive agent to be attached to, adsorbed onto the NPC surface, or trapped (dissolved or disseminated) inside the matrix. In addition, the preparation method employed can produce two different structures: nanospheres and nanocapsules. They will impact the NPC’s bioactive release profile, average size, zeta potential, manufacturing yield, loading capacity, and entrapment efficiency [47]. 

The extraordinary pharmacokinetic control displayed by polymeric nanoparticles enables them to encapsulate and distribute various medications while successfully modifying their outer layer through chemical processes. One of the most well-known NPs in this context is polylactic-co-glycolic acid polymer composite; other NPs include gelatin, albumin, chitosan, and pectin, along with poly(lactic-co-glycolic acid) (PLGA) [16]. Figure 4 depicts the nanocarrier system-mediated therapeutic effects of GA.

Nanoencapsulation (NEC) has been extensively utilized in drug delivery systems for various purposes. The most widely used techniques for encapsulating drugs are polymerization, reverse micellar, ultrasonication, high-pressure homogenization, emulsification, solvent evaporation, etc. Figure 5A,B show a typical development process for vesicular carrier systems and polymeric core–shell nanocarriers, respectively. NECs are methods for producing lipidic and polymeric nanocarriers by superfinely coating drug particles with inert excipients or polymers. An encapsulated system is a particle surrounded by a matrix or reservoir. It can provide much more flexibility for site-specific drug delivery because of its variable physical and surface chemical features of bio-mimicked matrix composition. The schematic below explains some frequently employed methods for encasing bioactive compounds [14,48].

##### Liposome

A liposome is a bilayer lipidic vascular carrier system that holds cholesterol and phospholipids. Its size ranges from 25 to 2.5 nm. Their obvious advantages are their ability to contain a broad spectrum of materials and their structural versatility. Liposomal encapsulation allows for the entrapment of drugs at the phospholipid bilayer’s aqueous core or at the bilayer interface, despite the drugs’ widely different lipophilicity and solubility. Natural lipid-based liposomes exhibit minimal intrinsic toxicity and are biodegradable, immunogenicity-free, and physiologically inert. As such, it is expected that drugs encapsulated in liposomes can be administered without rapidly deteriorating, leading to the most minor side effects. Many herbal remedies are blended with liposomes to create targeted pharmaceuticals for illness prevention or treatment [49]. The liquid enclosing the cavity or the layers of the membrane dissolves the biologically active biomaterial in liposomes [50]. During a comparative study, pure gallic acid has shown various values for their pharmacokinetic parameters, like C_max_: 2.5 ± 0.18 µg/mL, AUC_0–t_: 2.82 ± 0.2 µg·h/mL, AUC_0–∞_: 3.09 ± 0.21 µg·h/mL, T_max_: 1.75 ± 0.78 h, and liposomal formulation’s pharmacokinetic parameter values are as follows: C_max_: 5.57 ± 0.48 μg mL^−1^, AUC_0–t_: 12.45 ± 0.93 μg·mL^−1^ h, AUC_0–∞_: 13.33 ± 0.87 μg·mL^−1^ h, improved T_1/2eli_: 2.12 ± 0.18 h, which depicts that liposomal formulation of GA has better pharmacokinetic properties than pure GA [51]. 

Andrade et al. have formulated GA-loaded transferrin-functionalized liposomes to manage Alzheimer’s disease. The initial step in creating GA-loaded liposomes was to use various techniques to determine which formulation had the best physicochemical properties, including size, PDI, zeta potential, and EE. It was revealed that the liposomal composition generated through the reverse-phase evaporation approach was the best fit over brain redistribution regarding the biophysical features of NPs. Transferrin was subsequently added to the liposomes’ surface to functionalize their ability to cross the BBB after loading them with GA [52]. 

Altan et al. have prepared and evaluated the liposomal system of GA (5 mg) for wound healing activity in animal models. It was observed that GA–liposomes exhibited a significant wound-healing effect compared with free GA (Figure 6A). They improved bone regeneration in rats by increasing the expressions of bone morphogenetic protein-2 (BMP-2) and orthopantomogram (OPG). In contrast, the nuclear factor-κB ligand-receptor activator (RANKL) expression and inflammation were lowered comparatively [53]. 

##### Solid Lipid Nanocarrier

Solid lipids with dimensions ranging from 50 to 1000 nm are called SLNs, also known as lipospheres or solid lipid nanospheres. Wax, fatty acids mono, di, and triglycerides, and combinations of these can all be utilized to generate them. SLNs can be produced on a large scale by replacing liquid lipids (oil) in an emulsion of water and oil with a solid lipid using several commercially viable methods [54]. Cordova et al. reported that GA–SLNs (designated as G8-NVM) have exhibited significant anti-cancer effects against B16F10 cell-induced pulmonary cancer. In the metastasis model, GA derivative-SLNs produced by heat homogenization have shown anticancer activity. In vivo, metastasis was observed following the administration of GA and G8-NVM to Swiss albino mice engineered to develop metastases. Further, data show fewer toxicological factors (weight variations, hematological and biochemical characteristics, histological examination, and oxidative state in the liver) when G8-NVM is compared with a free drug [55]. 

##### Nanoemulsion

Nanoemulsions (NEs) typically have diameters between 10 and 100 nm. In contrast, nanoemulsion has distinct rheological properties that prevent phase separation, making it noticeably more stable than conventional emulsion. Researchers assessed the physical characteristics of simple emulsions and nanoemulsions [56]. Nanoemulsion can be prepared into various dose forms, including liquids, creams, aerosols, gels, and foams. NEs have been administered through many routes, such as the skin, through the mouth, veins, intranasally, pulmonary, or ocularly. In the cosmetic and pesticide industries, they are used as an aqueous basis for organic delivery due to their superior ability to dissolve over basic dispersions of micellar matter and their improved kinetic stability compared with rough emulsion. Their small droplet size directly influences their long-term physical stability by lessening the capacity of conventional destabilization processes, including creaming, sedimentation, and coalescence [57]. 

Costa et al. have developed gallic acid nanoemulsion and emulsion. They observed that nanoemulsion demonstrated increased oxidative stability due to smaller droplet size in contrast to coarse emulsion. It implies that, compared with what we initially believed, the reactant concentrations—rather than the reaction surface—determine the rates at which chemical reactions occur during oxidation. As the water/oil ratio arose, emulsions overall oxidative stability improved [58]. 

Mohamad et al. have prepared GA-loaded niosome-based NEs (GANE) using the green synthesis method. They have evaluated the lung anticancer effect of GANE against A549 cell lines and in animal models. GANE has shown significant anticancer and antifibrotic effects against lipopolysaccharide (LPS)-mediated fibrosis in rats at 32.8 and 82 mg/kg b.w. GANE has increased lung endogenous antioxidant levels of GSH, GPx, and SOD production. At the same time, the MDA level dropped. Dexamethasone and GANE increased the synthesis of IL-4 while decreasing TNF-α and IL-6. However, in contrast to the rats treated with LPS, the lung p38MAPK, TGF-β1, and NF-κB gene expression was downregulated in the rats given GANE. GANE can restrict the migration and proliferation of lung fibrotic cells by inhibiting lipid peroxidation, oxidative stress, and cytokines, as well as downregulating the expression of the genes for TGF-β1, NF-κB, and p38MAPK (Figure 6B) [59]. 

##### Transfersomes

Transfersomes, often referred to as elastic vesicles, are extremely deformable and can only carry therapeutic drugs when administered in nonocclusive settings. They can pass through undamaged skin when subjected to a hydration gradient. As “edge activators”, the surfactants give the transfersome’s structure exceptional deformability, which facilitates its passage through stratum corneum (SC) pores [49]. 

Transfersomes can store a range of pharmacological substances with different solubilities since they consist of both hydrophobic and hydrophilic moieties. They do not suffer too much whenever they twist and squeeze through apertures that are between five and ten times lower than their own diameter. Like liposomes, they are biocompatible and biodegradable since they are made of natural phospholipids. They have high entrapment efficiency and can capture over 90% of lipophilic medicines. For example, they protect the medication within a capsule from proteins and peptides that undergo metabolic breakdown [60,61]. 

Pereira-Leite and Ventura have developed and optimized GA-loaded transfersomes using a Box-Behnken factorial design. It was created using a thin film hydration technique, using soy lecithin and tween 80 as excipients. The optimized formulation has shown desirable particle size (107 nm), polydispersity index (PDI) (0.20), and loading capacity (0.73%) for transdermal delivery [62]. Moreover, Wongrakpanich et al. have formulated the GA-containing transfersomal system of *Phyllanthus emblica* extract for topical delivery, especially for hair growth promotion. PE-loaded transfersomes had a particle size of 228 nm and a PDI of 0.25, respectively. To create follicles, it can stimulate the expression of the hair growth gene. Therefore, PE-loaded transfersomes are a potentially effective delivery method for targeting hair follicles to encourage growth [63]. 

##### Niosomes

Niosomes are vesicular nanocarriers that self-assemble and are produced by hydrating synthetic surfactants with suitable concentrations of amphiphilic substances, such as cholesterol. Like liposomes, niosomes can be unilamellar or multilamellar, transport hydrophilic and lipophilic medicines, and deliver pharmaceuticals to their intended location. Additionally, niosomal vesicles, often non-toxic, have fewer production costs and exhibit longer stability under various conditions, thereby mitigating some of the disadvantages associated with liposomes. In particular, size, shape, and surface chemistry determine the niosome features, which can alter the drug’s intrinsic kinetics and ultimate drug targeting to the pathological regions [64]. 

Manosroi et al. have developed a *Terminalia chebula*-containing GA-loaded niosomal-gel system for topical delivery. The highest cumulative amount was seen in the pure GA (GS) gel after 12 h, at 898.98 ± 18.96, 21.87 ± 1.00, and 15.89 ± 0.75 µg/cm^2^. The corresponding values for stratum corneum, epidermis, and dermis layer and the receiving solutions were 35.96, 0.87, and 0.64%, respectively. With a 12 h GA exposure, the semi-purified fraction (SS) gel showed the lowest cumulative amount at 1.12 ± 0.86 and 0.55 ± 0.17 µg/cm^2^, which equated to 0.19 and 0.09% in the receiving solution and epidermis and dermis layer, respectively. Regarding stratum corneum, the portion loaded in elastic niosomes (SE) (32.92%) and nonelastic niosomes (SN) (33.9%) gels had lower gallic acid percentages than the semi-purified fraction in gel (SS), which was 41.01% higher. Pure GA-based niosomal gel penetrates the skin more efficiently than the other forms. Therefore, this formulation is applicable for topical administration and unsuitable for transdermal application because the drug could not cross the stratum corneum layer to reach the systemic circulation [65]. 

##### Micro and Nanoparticulate Carrier Systems

Small, spherical particles called microbeads are employed in drug delivery systems to transfer drugs to the body efficiently and sustain the right concentration of the medicine. They are components of several unit dosage forms, each with desired qualities, like micro granules, pellets, and microcapsules. Because microbeads can deliver drugs with targeted and controlled release, they are an advantageous option compared with single-unit dose forms. Ionotropic gelation techniques, such as emulsion gelation, polyelectrolyte complexation, and internal and external gelation, are commonly used to manufacture microbeads for drug delivery systems. These techniques are the best for microbead preparation because they do not use strong chemicals or high temperatures [66]. 

In an investigation by the author, the electrospray method was employed to generate alginate beads ranging from approximately 200 µm to 1.3 mm. This was achieved by modifying the working voltage and the alginate beads’ capacity to encapsulate GA, a hydrophilic phenolic compound model. Between 7 and 12 g/100 g of GA were loaded into each alginate bead sample. The results of differential scanning calorimetry and Fourier transform infrared spectroscopy confirmed the incorporation of GA. Furthermore, none of the formed alginate beads showed signs of GA autoxidation. The release profile results showed that the release in gastric fluid was slower than that in simulated intestinal fluid. The GA loading amount and the alginate beads’ size impacted the release pattern. According to the findings, the electrospray approach of creating alginate beads was a potentially effective way to distribute water-soluble GA [67]. 

Microspheres are systems that administer medications in many particles using natural and artificial materials. The stability, bioavailability, and ability of the medication to target a particular area at a predetermined rate are all increased using microspheres. Different microspheres include bioadhesive, polymeric, radioactive, biodegradable, and floating types. Microspheres are utilized explicitly in novel drug delivery systems [68]. Researchers have prepared GA-containing microspheres with sodium alginate (NaAlg). The effect of microencapsulation on the antioxidant activity of GA was investigated by measuring the 2,2-diphenyl-1-picrylhydrazyl (DPPH) radical scavenging capacity, and it was shown that GA in the microspheres retained antioxidant activity. Anticancer activity was performed by 3-(4,5-dimethylthiazol-2-yl)-2,5-diphenyltetrazolium bromide (MTT) analysis on the Caco-2 cell line. As a result of the study, it was observed that the anticancer activity was maintained depending on the dose and time. Microencapsulation has been shown to preserve the antioxidant and anticancer activities of GA [69]. 

Nanocapsules are a type of nanoparticle consisting of a protective shell containing bioactive materials and a core. Its many advantages are reduced drug toxicity, enhanced bioavailability of medicinal compounds, and prolonged drug release. Nanocapsules are tiny particles ranging from 10 nm to 1000 nm [70]. Green tea, witch hazel, sumac, and oak bark are among the herbs that contain gallic acid, a naturally occurring antioxidant that contains polyphenolic components. It finds widespread application in foods, medications, and cosmetics. Researchers are interested in GA due to its anti-inflammatory, antioxidant, antimutagenic, and anticarcinogenic qualities.

Mu et al. formulated bovine serum albumin (BSA)-based GA–Fe^3+^ nanoparticles to treat tumors in BALB/c nude mice. The prepared iron–NPs of GA had shown 3.5 nm particle size and good biocompatibility. In an in vivo study, ultrasmall GA–Fe@BSA NPs were quickly removed by renal filtration and avoided phagocytosis in the reticuloendothelial system. Additionally, this feature lessened the possibility of toxicity brought on by prolonged bodily retention of the injected drugs. Significant alterations in T1-weighted MRI signal were noted in individuals harboring tumors before and after intratumor injection with GA–Fe@BSA, which effectively eliminated solid tumors concurrently treated with laser ablation. These findings indicate a promising future for GA–Fe@BSA NPs as effective and safe theranostic clinical use. Thus, the present investigation suggested that high-performance GA–Fe@BSA NPs may be useful as theranostic for managing tumors (Figure 7) [71]. 

In another study, Aldawsari et al. developed PLGA-based GA–NPs for transdermal delivery. The outcomes show that this approach produced homogenous, precisely spherical, negatively charged particles appropriate for application as skin patches or ointments. Subsequent investigation reveals that these GA@PLGANPs are excellent candidates for various applications due to their strong antibacterial and antioxidant properties against multiple microbes. Furthermore, by adversely affecting bacterial cell walls or membranes, these nanoparticles have been shown to drastically suppress microbial growth and effectively reduce oxidative stress [72]. 

In a study, Dehghani et al. examined the potential preventive benefits of pure GA and GA–NPs (nanogallic acid) against cisplatin-induced nephrotoxicity in rats. They reported that the nanogallic acid groups showed a significant decrease in the formation of mitochondrial ROS, mitochondrial membrane damage (ΔΨm), mitochondrial MDA, TNF-α, and IL-6. In addition, the nanogallic acid-treated groups showed a significant increase in mitochondrial GSH, mitochondrial superoxide dismutase (MnSOD), mitochondrial GPx, and mitochondrial catalase. A histopathology study reported that the cisplatin-induced group shows proximal tubule damage, infiltration of inflammatory cells and red blood cells (RBCs), and accumulation in the kidney. However, the histological criteria were lessened in the groups treated with gallic acid (10, 50, and 100 mg/kg) and nanogallic acid (10 mg/kg) compared with the cisplatin group. Thus, they concluded that the nanogallic acid is more efficient than pure GA in treating renal complications (Figure 8) [73]. 

##### Micelles

Drugs can be delivered in aqueous solutions using micelles, naturally occurring nanoscale aggregations. The structure comprises amphiphilic molecules, with the hydrophobic segments clumping in the core and the hydrophilic segments creating the outer shell. Many variables affect polymeric micelles’ stability, including the surrounding environment, drug encapsulation, polymer composition, and other thermodynamic and kinetic properties [74]. As a result, systems that are distinct from other nanocarriers are created. These include smaller sizes that allow for passive delivery to solid malignancies, even those that are not well soluble, more effective cellular internalization, improved solubilization of substances that are hydrophobic incorporated in the lipophilic core, and longer blood circulation times due to the naturally occurring hydrophilic corona [75]. 

In a study, Basir et al. prepared two kinds of polymeric micelles (PMs) such as GA and naringenin (NAR) that were made using the solvent casting process with varying amounts of polyethylene glycol (PEG) and D-α-tocopheryl polyethylene glycol 1000 succinate (TPGS). NAR–PEG–TPGS PMs had a particle size of less than 30 nm, whereas GA–PEG–TPGS PMs had a more extensive particle size range of 171–205 nm. The larger particle sizes were linked to higher TPGS amounts in GA–PM, but smaller particles were connected to greater TPGS concentrations in NAR-PM. The range of PDI values for these drug-loaded PMs was 0.32 to 0.74. FTIR data showed that all the PM samples possessed stretching vibrations, including O–H and C=O. Ultimately, NAR–PEG–TPGS PMs outperformed GA–PEG–TPGS PMs regarding relevant physicochemical properties for the administration of cancer nanodrugs [76]. 

Radwan et al. formulated micelle-loaded lipid nanocapsules (RMLNC) utilizing the Box-Behnken design to treat hepatic fibrosis. The developed micelle displayed a sustained release profile after two hours of burst GA release. During in vivo biodistribution imaging, rat livers were the primary organ where rhodamine-B-loaded RMLNCs were concentrated. Compared with GA, GA–RMLNC demonstrated increased anti-proliferative activity, efficient internalization into activated hepatic stellate cells, a notable down-regulation in the expressions of pro-fibrogenic biomarkers, and increased apoptosis in activated hepatic stellate cells [77]. 

##### Dendrimer

Three-dimensional, hyperbranched globular nanopolymeric formations are called dendrimers. Their attractive features include nanoscopic size, limited polydispersity index, greater control over molecular structure, and availability of multiple functional groups at the periphery, as well as inner cavities. The ability to quickly synthesize novel dendritic carriers has been made possible by advancements in synthetic chemistry and characterization techniques. Furthermore, a wide range of dendritic scaffolds with various nanoscopic diameters and many functional end groups are accessible [78]. Gallic acid dendrimer (GAD), which functions as a more effective radical scavenger than its pure form, was highly efficient in preventing the auto-oxidative degradation of essential oils. In particular, GAD has shown the potential to significantly slow down the degradation during the early stages of thermally triggered oxidative degradation. It contrasts with free GA, which exhibits discrete activity during the early stages of degradation but functions more as a pro-oxidant than a preservative over extended periods. Once it reaches a degradation entity plateau, GAD maintains it indefinitely or even initiates a decline, exhibiting a radical scavenger action faster than peroxide formation. To generally prevent the degradation of essential oils or lipid substrates, GAD is advised as a cutting-edge semi-synthetic polymeric substitute for the current minor molecules-based preservative additives [79]. Priyadarshi et al. have developed a GA-loaded polyamidoamine (PAMAM) dendrimer for treating human colon cancer. They found that GA–dendrimer has shown a profound effect against HCT116 cells. GA–dendrimers have shown a therapeutic effect through arresting inflammatory responses, cell migration, and cell proliferation to augment apoptotic cell death in cancer cell lines. The findings demonstrate that the PAMAM–GA conjugate suppresses the growth of cancer cells originating from various sources, enhances the uptake of GA by cells, hinders colonogenic potential, and limits the migration of cancer cells by downregulating matrix metalloproteinase-9 (MMP-9) expression. It prevents pro-inflammatory cytokines release and nuclear factor-κB (NF-κB) activation, which causes apoptotic cell death in HCT116 cells as opposed to necrosis [80]. In addition, many researchers have designed and evaluated GA-loaded dendrimers to enhance their permeability and therapeutic efficacy [81,82,83]. 

##### Hydrogels

Hydrogels, three-dimensional (3D) networks of crosslinked, highly hydrophilic polymers, can absorb large amounts of water and biological fluids. Hydrogels with different properties have been developed based on synthetic and natural polymers and their combinations for tissue engineering, drug delivery, and water purification, among other applications. Water-swollen hydrogels are soft polymers that have the potential to mimic the mechanical properties of animal tissues, and they are highly beneficial for regenerative medicine. Hydrogels have also been used in many different contexts, including scaffolds, injection-based hydrogels, nanogels (i.e., hydrogel nanoparticles), nanofibers, microgels, and microspheres, and very thin films known as hydrogel membranes (HMs) [84]. 

Using a straightforward and efficient technique, the author created a self-adhesive hydrogel inspired by mussels for strain-controlled transdermal administration of drugs in monoliths in a study. GA is used to alter alginate in manufacturing, and then polyacrylic acid is polymerized in situ. Because of the many hydrogen bonds, the resultant GA–hydrogel is an intensely entangled and interpenetrating supramolecular network. It sticks firmly to metal, glass, and polymers in wet and dry environments and is incredibly stretchable (800% strain). It also has a very porous and layered interior structure, giving it remarkable flexibility and effective drug administration. According to the results of the kinetic experiment, under 100%, 50%, 25%, and 0% tensile strains, 77.47%, 82.09%, 87.64%, and 42.54% of the already loaded caffeine in GA get released within an hour [85]. 

Recently, Li and co-researchers have developed liposome–gelatin-based hydrogels of GA for the treatment of osteoarthritis (OA). These novel gel systems of GA have exhibited remarkable cytotoxic effects against OA through the PI3K/AKT signaling pathway and by promoting IL-1β-mediated ATDC5 cell proliferation (Figure 9). Hydrogels can regenerate the damaged cartilage and increase the generation of glycosaminoglycans in OA animal models. The developed GA–hydrogels have shown antiarthritic effects in a sustained and controlled manner. Thereby, GA–hydrogels have become promising nanocarriers for the management of OA [86]. 

In this sequence, Yu et al. have prepared GA hydrogel by free radical polymerization using chitosan and 2-acrylamido-2-methylpropane sulfonic acid (AMPS). It was found that, after 48 h, the highest release of GA occurred at pH 1.2 (85.27%) as opposed to pH 7.4 (75.19%). The developed hydrogels were shown to be biodegradable (8.6% degradation/week), possess antimicrobial properties against *P. aeruginosa*, *S. aureus*, and *E. coli*, and have antioxidant properties (73% DPPH and 70% ABTS). Thus, hydrogel is promising for transdermal drug delivery [87]. Table 1 provides a thorough overview of the different lipidic and polymeric nanoparticulate carriers of GA.

#### 3.3.2. Inorganic Nanoparticulate Carrier Systems of GA

Inorganic nanoparticles are defined as particles that are not based on carbon. Inorganic nanoparticles are generally perfect for cellular administration because of their many versatile properties. Their regulated release of the medications they contain, rich functionality, excellent biological compatibility, the potential for selective administration (e.g., killing cancer cells selectively while sparing unaffected tissues), and wide availability are some of these qualities. Because of their low toxicity and ability for controlled administration, inorganic nanoparticles present a fresh alternative to cationic and viral carriers [104]. 

##### Metal-Based Nanoparticles

Many metals, including zinc (Zn), copper (Cu), cobalt (Co), lead (Pb), silver (Ag), selenium (Se), palladium (Pd), cadmium (Cd), aluminum (Al), gold (Au), and iron (Fe), can produce metal-based nanoparticles (Figure 10). With their partially complete d-orbitals, transition metals are highly reactive materials that can produce metal-based nanoparticles. Metal-oriented inorganic nanoparticles are the most utilized kind and present good protection against antibiotic resistance. They work well against already resistant bacteria and specifically target several biomolecules that are hindered in the formation of resistant strains. Additionally, they use entirely different modes of action than those seen in traditional antibiotics [105]. The size range for metal-based nanoparticles is 10–100 nm. They can be shaped like spheres or cylindrical ones, among many other shapes. Elevated volumes to surface area ratios, pore widths, densities, and surface charges as well, amorphous and crystalline forms, high reactivity, and susceptibility to heat, humidity, sunshine, and air, among other characteristics, are a few of its peculiarities [16]. Various metal-based GA nanoparticulate carrier systems have been represented in Table 2.

##### Gold Nanoparticles

Red wine-colored gold nanoparticles are said to have antioxidant properties. Interparticle connections and the formation of networks of gold nanoparticles are the main factors influencing these nanoparticles’ properties. Gold nanoparticles are characterized by a wide range of dimensions and morphologies, from 1 nm to 8 m. The following shapes are among them: tetrahedral, irregularly shaped, icosahedral, decahedral, icosahedral, octahedral, hexagonal platelets, nanotriangles, nanoprisms, and icosahedral multiple twined. In biomedical science, gold nanoparticles are commonly utilized in photothermal therapy, immunochromatographic pathogen detection in clinical specimens, tissue or cancer imaging, and drug administration due to surface [106].

Colloidal nanoparticles of gold (AuNPs) are being studied as non-toxic drug delivery vehicles because of their advanced properties, which include a sizeable surface-to-volume proportion and the ability to change their charge, hydrophilicity, and activity through surface chemistries. Many biocompatible polymers have been used to adorn the surfaces of AuNPs to boost their stability, payload ability, and cellular intake. For example, Astra Zeneca and Cytimmune collaborated on AuNPs-based nanomedicine for cancer treatment. Aurimune (CYT-6091), a drug based on PEGylated (polyethylene glycol) colloidal gold particles, whose nontoxicity has been reported, is presently undergoing clinical studies. It can efficiently deliver tumor necrosis factor-alpha (TNF-α) to the sites of tumors. AuNPs are versatile agents with applications in combination with treatment and diagnostics (theragnostic), such as photothermal tumor ablation, radiation sensitization, and actual time in vivo/in vitro imaging [107]. 

Nanoparticles were established in multiple studies in comparison to the free compounds. In this study, GNPs–GA showed lower cytotoxicity than GA alone. Nevertheless, GNPs–GA were only harmful to cervical tumor cells at the exact high dosages; normal or Vero cells seemed unaffected. Additionally, the position and dispersion of GNPs–GA were tracked. GNPs–GA was found mainly in the cytoplasm and is taken up by endocytosis into the cells. Compared with control, GNPs–GA can increase cell proliferation at relatively low concentrations (20–80 mM). GNPs can, at specific dosages, promote the growth of keratinocyte cells [108]. Many functional groups can be connected on the particle surface of gold nanoparticles, including PEG, ssDNA, antibody, peptide, drug, fluorescence marker, and siRNA (Figure 11).

##### Silver Nanoparticles

Silver nanoparticles (AgNPs), highly promising and appealing nanomaterials, have drawn attention, particularly in drug delivery and biomedical applications. AgNPs are well known for their inherent antimicrobial properties. The capacity of AgNPs to function as nanocarriers for drugs has aided in the quick development of innovative treatments for various diseases, including cancers. The nanoparticle’s ongoing mechanisms related to the controlled dissolution, aggregation, and formation of oxygen radicals may also make them useful for novel pH-responsive systems for drug delivery in the current situation [109]. Ag–NPs have become a thrust area in the field of phytopharmaceuticals and herbal medicines for getting several pharmacological activities like anti-inflammatory, antimicrobial, anticancer, antidiabetic, wound healing, antiarthritic, anti-Alzheimer’s, Parkinson’s, etc. [110,111,112].

GA–silver nanoparticles (GA–AgNPs) are a type of inorganic nanocarrier that have been reported for both antimicrobial and antioxidant properties. AgNPs can be synthesized using gallic acid as a reducing agent, which converts silver ions into silver nanoparticles. GA–AgNPs exhibit antibacterial, antifungal, and antiviral properties, making them effective against various microorganisms. Gallic acid’s antioxidant properties enhance the stability and biocompatibility of AgNPs, reducing oxidative stress and inflammation. GA–AgNPs have potential uses in wound healing and tissue repair, antimicrobial coatings and surfaces, cancer treatment and imaging, drug delivery and release, and cosmetics and skincare products. The combination of GA–AgNPs offers improved biocompatibility, stability, and antimicrobial efficacy compared with traditional silver nanoparticles [113]. 

GA–AgNPs have been widely explored by researchers at the multidisciplinary platform. In this context, Nemčeková et al. have synthesized GA–AgNPs for bioanalytical study. They explored the electrochemical properties of Ag–NPs for controlling the drug release system at the targeted site, cancer [109]. 

Similarly, Majeed and the group have synthesized the GA–AgNPs for cancer targeting. They explored their anticancer potential against osteosarcoma cells (SAOS-2). Results stated that the GA–AgNPs-treated cancer cells exhibit classical apoptosis signs like membrane blebbing, chromatin condensation, and positive acridine orange and ethidium bromide (AO/EB) staining. GA–AgNPs have shown the mechanism of action against cancer cells via cell cycle arrest at the S and G2/M phases. They have confirmed the anticancer potential of GA–AgNPs against osteosarcoma through ROS-mediated pathways. Detailed information has been explained in Figure 12 [114]. 

Silver’s broad antimicrobial properties and potential anti-inflammatory and wound healing properties are raising significant interest in these composite materials as biomaterials. Silver nanoparticles can be used as substrates for surface-enhanced Raman scattering (SERS), another possible use in biomedicine. Silver nanoparticles can effectively improve visual stimulation due to their appropriate plasmon resonance frequencies and absorption properties. Silver nanoparticles are also used for metal-enhanced fluorescence (MEF) because they dramatically enhance fluorophore emission intensity. MEF is a powerful tool for fluorescent-based applications such as high-throughput screening, immunoassays, and macromolecular detection [115]. 

##### Palladium Nanoparticles

Palladium (Pd) is a rare, lustrous, precious metal with unique properties. Due to its specific characteristics, Pd is critical in producing catalytic converters, reducing harmful vehicle emissions. It is also used in various biomedical applications. It is a non-toxic and non-reactive element used in dental applications, especially for surgical instruments and medical implants [116]. 

Pd can be used to cover medical devices to lower the risk of infection because of its antimicrobial effect. It is a scaffold substance that promotes cell growth and the formation of new tissue in tissue engineering. It is also used in photodynamic therapy for cancer treatment. Pd is used in dental implants, crowns, and bridges because it is biocompatible, corrosion-resistant, and allowable with other metals. Pd-based coatings and dressings can accelerate tissue regeneration, inhibit bacterial development, and improve wound healing. Because of its electrocatalytic solid activity, Pd is employed in biosensors to identify biomolecules like glucose. Because Pd can influence brain activity, it treats neurological disorders, including Alzheimer’s and Parkinson’s disease. Moreover, Pd is utilized in drug delivery systems, like nanoparticles, to release drugs at the targeted sites [116,117]. 

GA-based palladium nanoparticles (GA–PdNPs) are nanomaterial types with unique features and potential applications. GA, a polyphenolic compound found in plants, is employed in preparing Pd–NPs due to its stabilizing and reducing activity. GA molecules coat the Pd–NPs, preventing them from aggregating and providing stability. These nanoparticles have shown promise in various fields, including catalysis, biomedical applications, environmental remediation, and electronics. The synthesis of GA–PdNPs typically involves a simple, eco-friendly, and cost-effective method, making them an attractive option for various industrial and research applications [118]. 

Karimi et al. have synthesized the GA–PdNPs for antioxidant and antimicrobial activity. Researchers have used plant extracts (Hibiscus sabdariffa) containing GA for PdNP synthesis. Further, they coat the PdNPs using Ag and synthesize them as Ag–Pd–NPs. Ag–Pd–NPs have shown strong antibacterial action against several pathogenic microorganisms that cause a variety of human diseases, including *Candida albicans*, *Pseudomonas aeruginosa*, *Escherichia coli*, *Enterococcus faecalis*, and *Bacillus subtilis* [119]. GA–PdNPs have been synthesized for catalytic activity by Mondal et al. [120]. 

##### Silica Nanoparticles

Silica–NPs (Si–NPs) have become interesting nanomaterials that have various pharmaceutical uses. Mesoporous Si–NPs have been explored widely in recent years due to their significant characteristics. Its specific properties, like colloidal stability, make it more effective in drug targeting due to its high transparency and cell absorption. Modified surface and biocompatibility are the advantages of mesoporous Si–NPs, making it a promising material for a wide range of uses, including photochemistry, optics, biomedicine, adsorption, separation, and storage [121]. 

Hu et al. have developed GA–Si–NPs using SiO_2_, which was chemically bonded to silica nanoparticles to control the release of GA by the hydrolysis of the chemical bonds. A modified Stober approach was utilized to synthesize GA–Si–NPs. The prepared GA–Si–NPs have 89.39% drug loading efficiency and a particle size of 30 nm. In vitro release investigation showed that GA gradually released from the GA–Si–NPs. Furthermore, the GA’s antioxidant capacity increases slowly over the immersion period, suggesting that it would be a highly effective antioxidant for scavenging 1,1-diphenyl-2-picrylhydrazyl (DPPH) radicals over an extended-release period. The work offered a novel approach to delivering GA that has sustained antioxidant efficacy and controlled release. [122]. In another study, Szewczyk et al. studied the physiochemical properties of different antioxidants like GA, protocatechuic acid (PCA), chlorogenic acid (CGA), and 4-hydroxybenzoic acid (4-HBA), which were loaded in mesoporous Si–NPs. Here, GA-loaded mesoporous silica showed the highest absorption and antioxidant properties. Thus, they concluded that GA-loaded mesoporous Si–NPs preserved the highest antioxidant capacity among all investigated compounds [123]. In a study, Fahmy et al. assess the antidepressant effectiveness of free GA and GA–Si–NPs in a rat model of reserpine-induced depression. The result showed that the free GA was more successful in raising levels in the cortex, hippocampus, and hypothalamus. In contrast, GA–Si–NPs were more successful in raising serotonin levels in the striatum. The four brain regions under investigation saw an increase in neprilysin response to GA–Si–NPs. Compared with the depressed untreated group, free GA raised dopamine levels in the cortex and striatum, while GA–Si–NPs boosted dopamine levels in the hippocampus and hypothalamus [124]. In another study, Petrisor et al. used a vacuum-assisted technology to load GA into the pores of two mesoporous materials. They measured the release of the GA and its biological activity. The mesoporous structures based on GA have shown efficacy in mitigating the inflammatory response and have potential applications in microbiota restoration [125]. 

##### Iron Nanoparticles

Iron nanoparticles (Fe–NPs) are tiny particles of iron that are typically between 1–100 nm in size. Fe–NPs, among other nanomaterials, have drawn more attention due to the small size of their particles, surface characteristics, low toxicity, high magnetic, and wide range of scientific uses [126]. Nowadays, there is a lot of interest in magnetic nanoparticles in general, and iron is one of the most practical magnetic materials. Iron is a common material for magnetic recording media due to its magnetic characteristics [127]. Applications for Fe–NPs can be generally classified in several industries, including wastewater treatment, paint, textiles, medicine, food, and agriculture. Plant, microbial, and agro-waste product extracts are used in the biogenic generation of these nanoparticles, which lowers the cost and increases the synthesis efficiency [128]. 

GA–Fe–NPs are a type of nanoparticle that combines the benefits of GA with the exclusive properties of iron nanoparticles. GA has been shown to have anti-cancer properties, and when loaded onto Fe–NPs, it can be targeted to specific cancer cells, reducing side effects. The combination of GA and Fe–NPs has been shown to exhibit enhanced antimicrobial activity against certain bacteria and fungi. GA has neuroprotective properties, and when delivered using Fe–NPs, it may help treat neurodegenerative diseases like Alzheimer’s or Parkinson’s. Fe–NPs can deliver GA to specific sites in the body and reduce oxidative stress and inflammation. Fe–NPs can also be used as magnetic resonance imaging (MRI) contrast agents. The antioxidant and antimicrobial properties of GA–Fe–NPs may aid in wound healing and tissue repair. GA has been shown to have cardioprotective effects, and when delivered using iron nanoparticles, it may help prevent or treat cardiovascular diseases [129,130,131,132]. 

Dorniani et al. employed a sonochemical technique to synthesize magnetic Fe–NPs with a 1:2 molar ratio of Fe^2+^ to Fe^3+^ in an atmospheric condition. Further, the Fe–NPs were coated with GA and chitosan to create a core-shell structure. The iron oxide–chitosan–gallic acid (FCG) nanocarriers had a mean diameter of 13 nm. In contrast, the Fe_3_O_4_ nanoparticles had a spherical shape and a mean diameter of 11 nm, as revealed by transmission electron microscopy. The magnetic nanocarrier improved GA thermal stability. It was discovered that the active medication was released from the FCG nanocarrier in a controlled manner. Further, in a normal human fibroblast (3T3) line, the GA and FCG nanoparticles showed no toxicity and anticancer properties, but anticancer activity was higher in HT29 than in MCF7 cell lines [133]. 

Saleh et al. have prepared carboxymethyl chitosan-based GA iron oxide nanoparticles (GA–CTC–FeNPs) for cancer treatment. The cytotoxicity of GA–CTC–FeNPs was assessed both with and without bee venom to gauge their synergistic effects on the alveolar adenocarcinoma cell lines (A549 cells) and normal cell lines (WI-38 cells). Furthermore, the expression of apoptotic genes in A549 was assessed following treatment with GA nanoparticles. A crucial step in the cancer cell targeting procedure to reduce harm to normal cells is the ability of iron oxide–CMC–GA nanoparticles to specifically cause apoptosis in cancer cell lines more than in normal cell lines. GA–CTC–FeNPs have enhanced cytotoxicity and bee venom against A549 cells compared with WI-38 cells. Additionally, there was an increase in apoptotic gene expression [134]. 

##### Selenium Nanoparticles

Selenium nanoparticles (Se–NPs) are extremely small selenium particles, usually measuring between one and one hundred nanometers. Their distinct characteristics and prospective uses in numerous industries have piqued my curiosity. Se–NPs are utilized in drug delivery systems, imaging, and maybe as an agent for cancer treatment because of their capacity to target tissues or cells [135]. Se–NPs have been reported for their antioxidant effect, as they can scavenge free radicals and protect cells from oxidative stress. Se–NPs are employed in the biomedical field since they may be biocompatible [136,137]. Selenium nanoforms have garnered much interest lately due to their potent adsorbing capacity, huge surface area, high surface activities, high catalytic effectiveness, and low toxicity. Nanoselenium in different forms has demonstrated increased antioxidant and anticancer activity with minimal toxicity [138]. 

The possible synergistic effects of GA and selenium nanoparticles (GA–SeNPs) make this a fascinating field of research. Moreover, GA can function as a stabilizing agent to keep Se–NPs in the proper size and distribution by preventing them from aggregating. GA–SeNPs may be helpful in the treatment of diseases linked to oxidative stress or in the treatment of cancer due to their enhanced antioxidant qualities. Because of their combined qualities, the GA–SeNPs could be employed as medicinal agents, imaging agents, or drug delivery systems [139,140]. 

Lunkov et al. synthesize quaternary chitosan-modified Se–NPs covalently bonded with GA. The synthesized SeNPs were characterized by DLS, TEM, and AFM techniques. The SeNPs were stable with a size range of 119.5–238.6 nm and a positive charge of 34.86–46.73 mV. SeNPs show increased antibacterial and fungicidal activity. *C. albicans* (MIC 125 µg/mL) were considerably more susceptible to the modified Se–NPs toxicity. SeNPs synthesized at 55 °C showed the highest toxicity against the Colo357 and HaCaT cell lines among the nanoparticle samples [139].

In a study, Lv et al. outlined the ecologically friendly method of synthesizing GA–SeNPs by combining one mL of 50 mM GA with 10 mL of 30 mM selenious acid and one mL of 40 mM ascorbic acid, which served as the reduction reaction’s initiator. Following a 24 h reaction, the UV–Vis spectroscopic findings demonstrated the synthesis of GA–SeNPs. The pharmacokinetic studies of GA–SeNPs show an increased area under the curve (AUC) in the plasma concentration–time graph and a longer half-life than pure GA [141]. Table 2 shows various metal-based nanoparticles of gallic acid.

**Table 2 pharmaceutics-16-01202-t002:** Various metal-based nanoparticles of gallic acid.

S. No.	Nanocarriers	Physiochemical Characteristics	Outcomes	References
1	Gold nanoparticles	Av. Size = 40.8 ± 15.4 nm, zeta potential = –49.63 ± 2.11 mV, PDI = 0.141	GA–AuNPs repressed MMP-1 overexpression, oxidative stress, and type I collagen breakdown caused by high glucose.	[142]
2	Silver nanoparticles	Av. Size = 10–30 nm, zeta potential = −21.5	The produced nanoparticles boosted the expression of E-cadherin. They decreased EMT markers such as Vimentin, N-cadherin, and Snail-1, which may prevent cancer cells from developing the ability to spread radioresistance.	[143]
3	Silver nanoparticles	Av. size = 26.23 ± 9.92 nm	GC–AgNps exhibited potent antibacterial properties and toxic effects against *E. coli*.	[144]
4	Silver nanoparticles	Av. Size = 17.14 ± 1.25 nm, zeta potential = 29.1 mV	As determined by UV spectroscopy, the most stable compounds over time were QCG–AgNPs and QCG–AgNPs (T).	[145]
5	Palladium nanoparticles		It was discovered that a wide variety of functional groups were compatible with the documented reaction circumstances and that adding gallic acid affected the PdNPs’ size distribution and reaction rate.	[120]
6	Palladium nanoparticles		The as-synthesized GN@Pd–GA exhibits a remarkable ORR performance, nearly comparable to commercial Pt/C.	[146]
7	Palladium nanoparticles	Av. Size = 21 nm	Polyphenolics rich in the antioxidant gallic acid act as an effective reducing agent for Pt and Pd ions, which TEM, XRD, SAED, and ICP analysis proved.	[147]
8	Silica nanoparticles	Av. Size = 143 ± 12 nm, zeta potential = 41 ± 1.1 mV, %EE = 38.14%	According to the results, there was a noticeable difference in the extent of GA release between CS–MSNs and AP–MSNs. When compared with MCF-7 cells, CS–MSNs demonstrated superior killing potency.	[148]
9	Iron nanoparticles	Av. Size = 11 nm,	The antibacterial activity of IONP@GA varied depending on the strain of bacteria. Functionalized IONPs are thought to enter cells and cause cell wall damage by rupturing the β-1,4-glyosidic link, inhibiting bacterial growth.	[149]
10	Selenium nanoparticles	Av. Size = 5–10 µm	The author examined the antioxidant abilities of the nanocomposites using the DPPH radical scavenging assay, where the nanostructures were found to be highly potent.	[150]
11	Selenium nanoparticles	Av. Size = 162.6 nm, zeta potential = 6.73 ± 2.21 mV, λ_max_ = 247.5 nm	The most toxic to hepatocarcinoma cell lines (Colo 357) and keratinocytes (HaCaT) was found in QCGSeNPs 55 particles, which had an average diameter of 162.6 nm (DLS) and a zeta potential of 46.73 mV.	[139]

Av. Size: average size, PDI: polydispersity index, λ_max_: frequency, %EE: %entrapment efficiency.

## 4. Marketed Formulations of Gallic Acid

GA acid has been shown to have a wide range of beneficial effects, comprising anti-inflammatory, antimicrobial, antioxidant, cardioprotective, gastroprotective, and neuroprotective qualities [9]. GA belongs to BCS class III. Hence, permeability is the main thing stopping it from being absorbed [151]. When used topically, its limited bioavailability is one of its key drawbacks. Developing topical and transdermal routes of administration with nanocarriers may help increase the components’ limited bioavailability [152]. Oral GA is quickly absorbed and excreted, according to pharmacokinetic studies. However, structural alterations or dose form modifications are recommended to boost GA’s absorption. Fortunately, toxicology studies have shown that GA rarely shows overt toxicity or harmful effects in animal testing and clinical trials [10]. Several gallic acid nanocarrier techniques have been developed and tested to increase their bioavailability and effectiveness [16]. Table 3 shows different marketed formulations of gallic acid.

## 5. Patent on Gallic Acid and Its Derivatives

GA provides various health benefits like anti-inflammatory, anti-neoplastic, and anti-oxidant properties. It follows different mechanisms to counter other types of diseases. Researchers have developed different methods of preparation for gallic acid and its derivatives to make it reliable and easy to procure. Many of the researchers have patented their work; some of the developed methods are as follows—preparation of gallic acid by condensation and hydrolysis, extraction of gallic acid from tarapods, content, and procedures for combining pure gallic acid ester with proteins and peptides, etc. Some of the patents related to the preparation of gallic acid and its derivatives have been discussed in Table 4.

## 6. Clinical Trials Update on Gallic Acid 

Clinical trials are basically evaluative investigations involving human volunteers in prospective research studies to solve specific biological and behavioral therapy issues. It includes cutting-edge medications, vaccines, food options, nutritional supplements, medical technologies, and tried-and-true strategies that require additional investigation and analysis. Clinical trials provide safety, effectiveness, and dosage information. Depending on whether therapeutic approval is needed, they are put into practice only after being approved by the country’s ethics commission or health authorities. These authorities will decide the risk-to-benefit ratio of the trial.; their endorsement merely allows the experiment to continue and does not suggest that the medication is “safe” or “effective” [16]. The NIH, US website, https://clinicaltrials.gov/, has been visited to retrieve the information about GA and its clinical trials. A few clinical trials were reported on the website, represented in Table 5. 

## 7. Challenges and Future Perspectives of Gallic Acid Nanomedicine

Herbal, botanical, and phytopharmaceuticals have been used for centuries to prevent and treat various health conditions. With the growing interest in natural remedies, herbal medicine is gaining recognition for its pharmaceutical applications. Many herbal extracts and compounds are being studied for their potential to develop new drugs, and some have already been incorporated into modern medicine. For example, artemisinin from the Artemisia annua plant treats malaria, while willow bark extract containing salicylic acid is used to manufacture aspirin. Additionally, herbal medicine’s holistic approach to health and wellness aligns with the increasing focus on preventative care and personalized medicine, making it an exciting area of research and development in the pharmaceutical industry [153,154]. 

GA is a naturally occurring polyphenol from fruits, vegetables, and herbs. Numerous biological properties, including antimicrobial, anticancer, anti-inflammatory, and antioxidant, have been linked to GA. GA and its derivatives are used as food additives and supplements in the food industry. GA has been shown to be safe and effective in multiple studies, but its pharmacokinetic profile (PKs), such as low absorption, poor bioavailability, and quick elimination, limit its clinical use [9]. Fortunately, nanoformulation offers a way to improve these insufficient PKs of GA. Research has shown that enhanced pharmacokinetics and therapeutic effects can be achieved by employing NDDS. Another objective of nanocarriers was to strengthen the stability of a drug. Also, they need to be designed precisely to increase encapsulation efficiency and to allow controlled release at the target site. A drug’s lipophilicity and bioavailability can be increased by nanocarriers, which enhances the drug’s therapeutic efficacy. It suggests that using nanotechnology in medicine may improve therapeutic efficacy and produce the desired pharmacological response to support human treatment or recovery [155]. Undoubtedly, nanoparticle advancements could yield novel compounds derived from GA with noteworthy pharmacological potential. GA could potentially serve as a productive therapeutic agent for the treatment of several human diseases, such as melanoma, vaginal candidiasis, liver disease, depression, etc. [13]. 

In this regard, GA pharmacokinetic profiles were improved using different nanoformulation techniques. For example, research reports stated that phytosomes have improved hepatoprotective effectiveness and bioavailability of GA [156]. Elastic niosomes have been shown in another investigation to be particularly helpful in enhancing the stability and permeation of GA for topical anti-aging applications. The fact that medications are unable to reach the central nervous system (CNS), which is not first passed across the blood–brain barrier (BBB), complicates the therapy of neurodegenerative diseases (NDs). Nanomaterials can penetrate the BBB via both invasive and non-invasive methods. One method of delivering nanoparticles over the BBB is intracerebral or intracerebroventricular injection. Additional paracellular pathways consist of the following: cell-penetrating peptide (CPP), BBB crossing via the receptor, BBB crossing via cell-based strategy, BBB crossing by shuttle peptide-mediated BBB crossing, and intranasal administration technique [155]. In order to increase the phospholipid–GA complex’s bioavailability as well as retention in the human immune system, Bhattacharyya et al. aimed to create the gallic acid–phospholipid complex to boost its bioavailability and hepatoprotective effects. The complex’s morphology was described as rough on the surface and fluffy and porous. The complexation of GA with phospholipid was validated by FTIR, DSC, and XRPD data. An in vitro dissolution study revealed a controlled release profile of GA–phytosomes over 24 h, indicating that GA–phytosomes are crucial elements that permit the controlled release of GA [51]. 

Permeability of GA was utilized with both niosomes as opposed to pure GA, according to Manosroi et al. (2011). When comparing elastic niosomes to non-elastic niosomes, there was better GA permeability. The architectures of both unilamellar and multilamellar elastic and nonelastic niosomes loaded with a pure drug and a drug proportion that was semi-purified showed increased stability of the niosomes. Therefore, for topical anti-aging applications, niosomes are incredibly elastic and valuable in expanding the drug’s strength and permeability [65]. Shafiee et al. have demonstrated the anti-tumor effect of GA using graphene oxide (GO) as a nanoparticle. A phosphate buffered saline solution was used to evaluate the resulting nanoformulation’s in vitro release profile (GOGA), which showed sustained release of GA and consequent improvement in therapeutic efficacy and reduction in dosing frequency. Following a 72 h treatment of varied quantities of GA, GO, and GOGA on liver cancer cells (HePG2 and 3T3), in vitro biological experiments demonstrated that GOGA exhibited the highest suppression of malignant proliferation of cells without interfering with the expected proliferation of cells [157]. Because of its pharmacological activity, gallic acid has several health benefits. However, despite these advantages, its limited utility is brought on by poor absorption, insufficient bioavailability, rapid metabolism, and a high clearance rate. Therefore, there are continuous quests on nanotechnology and various NDDS for improving therapeutic efficacy and bioavailability of herbal bioactives like gallic acid. So that they could be used to prevent and treat several diseases with improved patient compliance [13].

Despite various materials being explored as co-delivery carriers, GA-based carrier-free NPs with exceptionally high chemotherapeutic drug content have emerged as a new trend in drug delivery research. Carrier-free nanoparticulate drug delivery systems offer distinct advantages, such as long circulation time, high tumor targeting efficiency, and controllable drug release [12]. 

Since nanoparticles have higher reactivity, efficacy, and bioavailability than conventional drug delivery systems, their use in clinical practice still needs improvement. Biocompatibility and immunogenicity are the main challenges for developing and implementing nanoparticle-based products. Hence, in this regard, regulatory bodies, such as the USFDA and the European Medicines Agency (EMA), have established guidelines for its toxicity, safety, quality, and efficacy profiles [158]. Research has demonstrated that the immune complement system’s activation in response to nanoparticularized drug delivery systems might result in hypersensitive reactions. One possible explanation for the hazardous response of nanoparticles could be the generation of free radicals, which induce oxidative stress. Free radicals can harm lipids, DNA, proteins, and other biological components when their concentration is high. In addition, nanoparticle-based products face strict regulatory approval processes before clinical applications [159]. 

Further, the high manufacturing costs may restrict the availability of nanoparticles. More research is needed to understand their compatibility with the human body. Thus, a standard set of protocols is essential for developing nanobased products for treating and curing diseases.

## 8. Conclusions

GA has several health advantages, including preventing and treating various diseases linked to inflammation and colorectal cancer. Although gallic acid has many benefits, it has several drawbacks, such as limited bioavailability and low permeability, making it difficult to absorb. Due to these limitations, studies on a different novel delivery system for GA have been conducted to enable its usage as a medication. Subsequently, several novel nanoformulations were the subject of intense investigation as potential effective carriers of this bioactive component. Various nanoformulations were created and investigated to identify the best gallic acid-loaded nanoformulations for treating and preventing various diseases. Thus, innovative formulation development has increased the bioavailability of this chemical component and generated a range of enriched therapeutic effects.

## Figures and Tables

**Figure 1 pharmaceutics-16-01202-f001:**
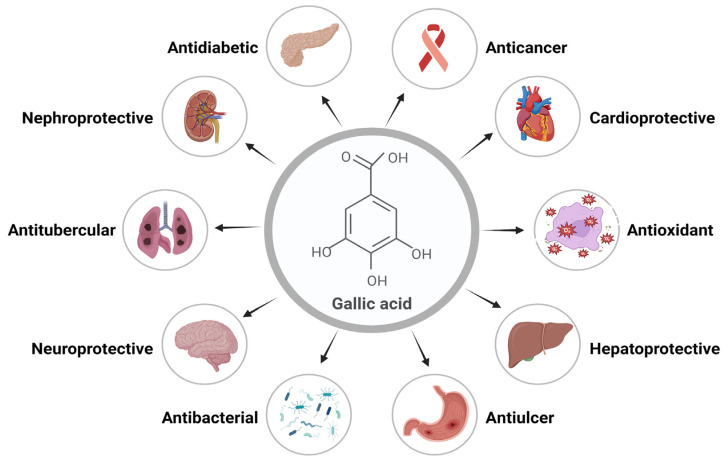
Gallic acid’s essential functions and biological activities.

**Figure 2 pharmaceutics-16-01202-f002:**
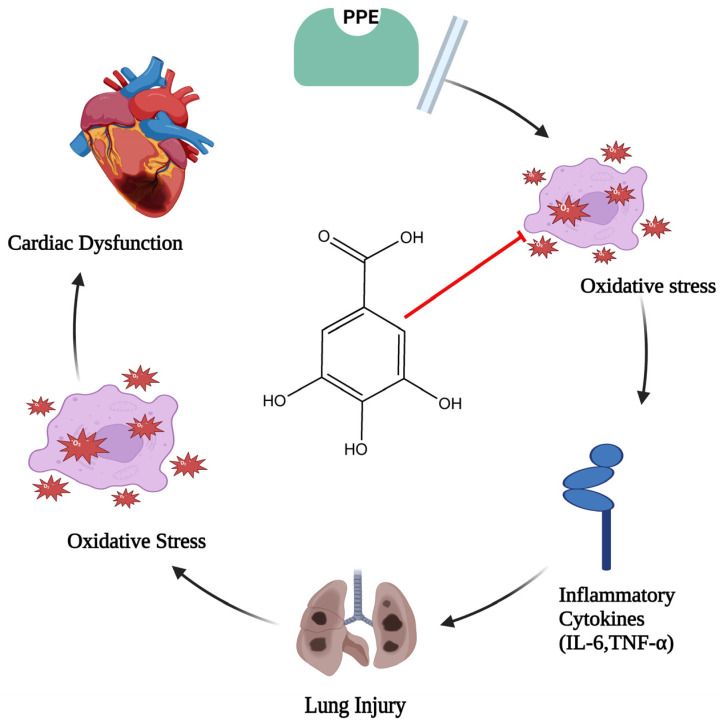
Diagram illustrating gallic acid’s defense against oxidative stress and inflammatory cascade (where the red arrow indicates inhibition of oxidative stress, which leads to lung injury and cardiac dysfunction).

**Figure 3 pharmaceutics-16-01202-f003:**
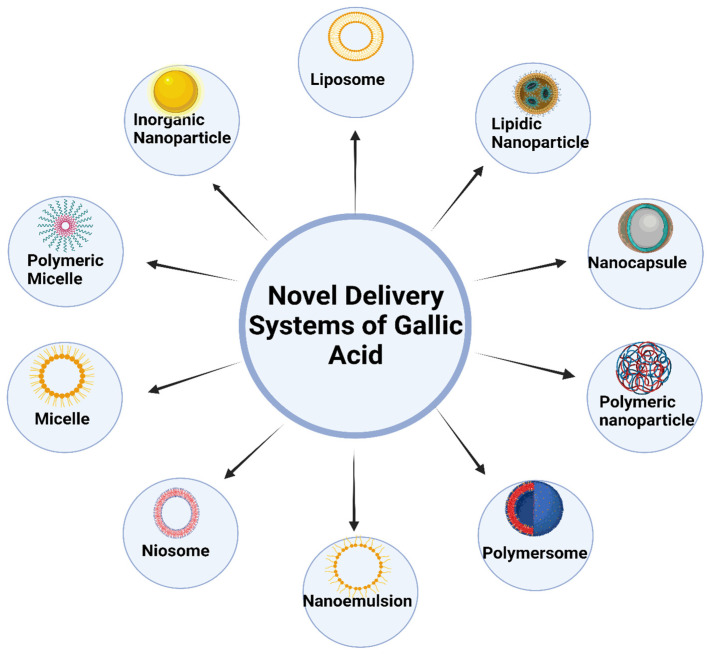
Various novel delivery systems of gallic acid.

**Figure 4 pharmaceutics-16-01202-f004:**
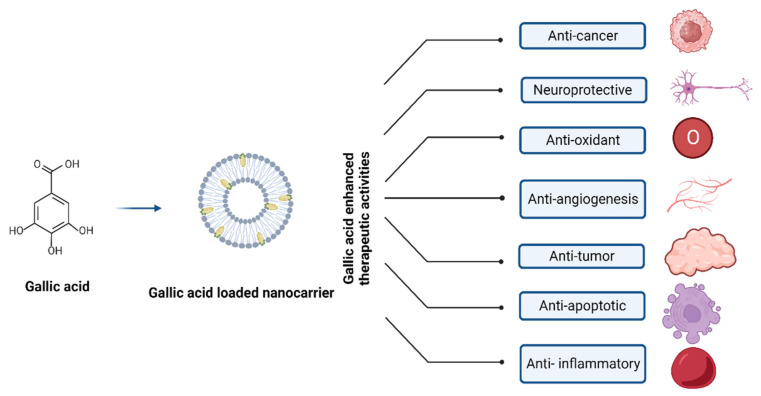
Nanocarrier system of gallic acid and their therapeutic effects.

**Figure 5 pharmaceutics-16-01202-f005:**
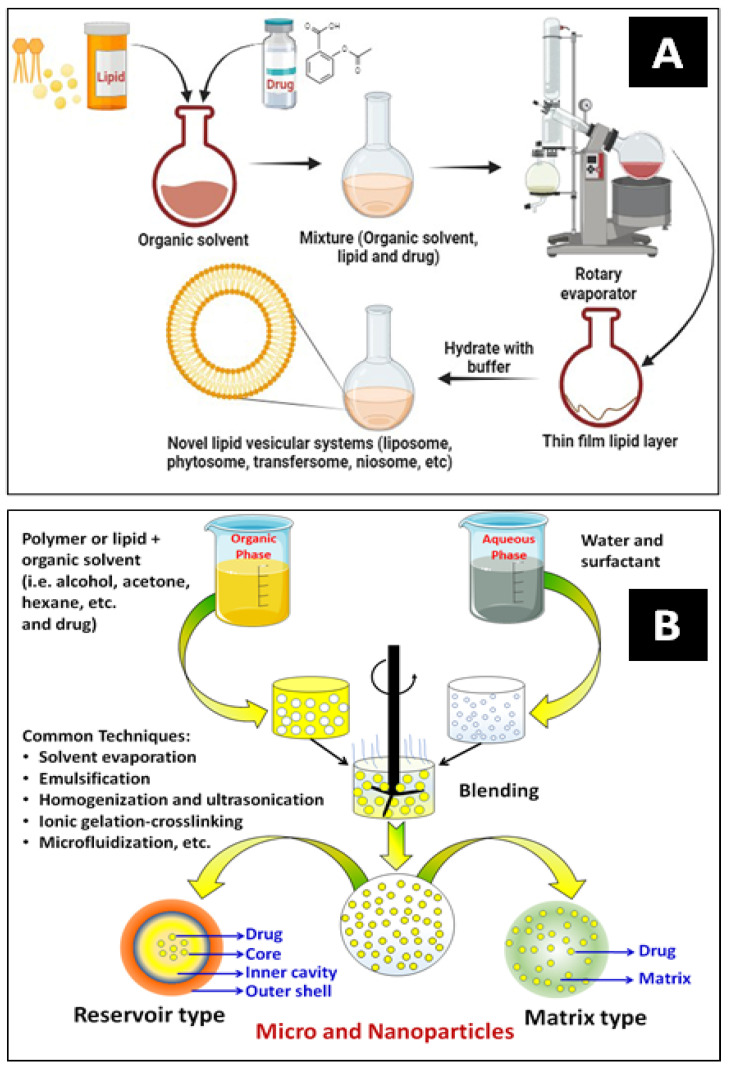
Schematics of lipidic (**A**) and polymeric nanocarrier (**B**) preparation methods.

**Figure 6 pharmaceutics-16-01202-f006:**
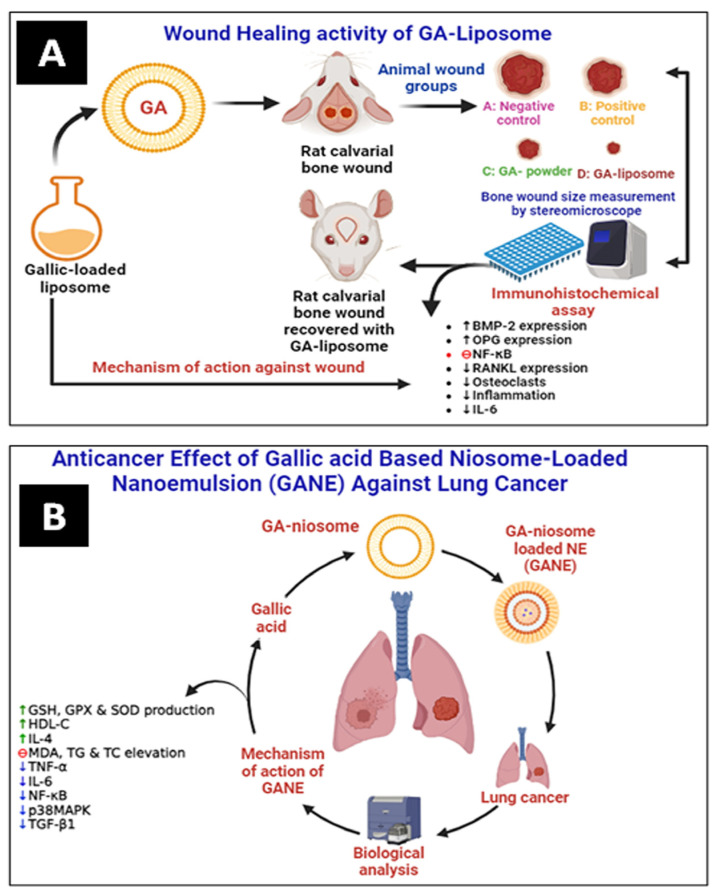
(**A**) Wound healing activity of GA–liposome against rat calvarial wound and (**B**) anticancer effect of GANE against LPS-mediated lung cancer (fibrosis) in the rat. GSH: glutathione, GPX: glutathione peroxidase, SOD: superoxide dismutase, HDL-C: high-density lipoprotein, TC: total cholesterol, and TG: triglyceride (↑ up regulation, ↓ down regulation).

**Figure 7 pharmaceutics-16-01202-f007:**
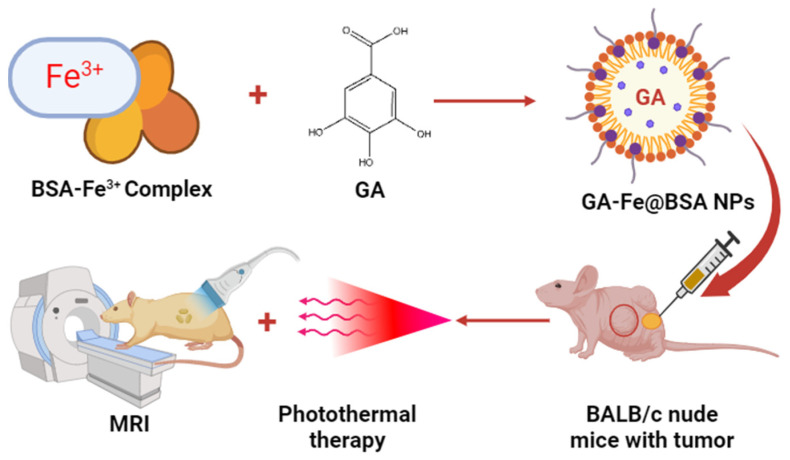
Antitumor effect of BSA-based GA–Fe nanoparticles in the mice.

**Figure 8 pharmaceutics-16-01202-f008:**
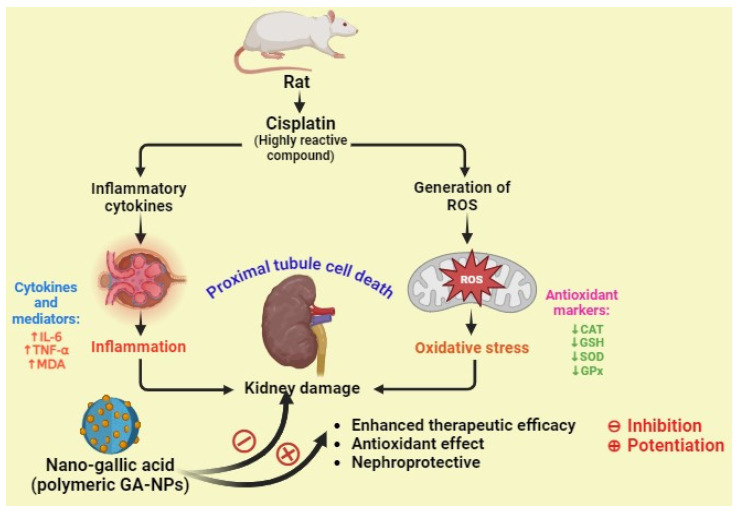
The nephroprotective activity of nanogallic acid against cisplatin-induced inflammation and oxidative stress in rats (↑ up regulation, ↓ down regulation).

**Figure 9 pharmaceutics-16-01202-f009:**
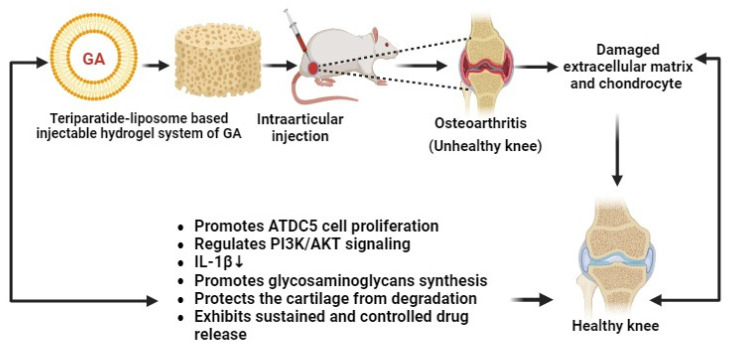
Teriparatide liposome-based injectable hydrogel system of GA for the treatment of osteoarthritis (↓ down regulation).

**Figure 10 pharmaceutics-16-01202-f010:**
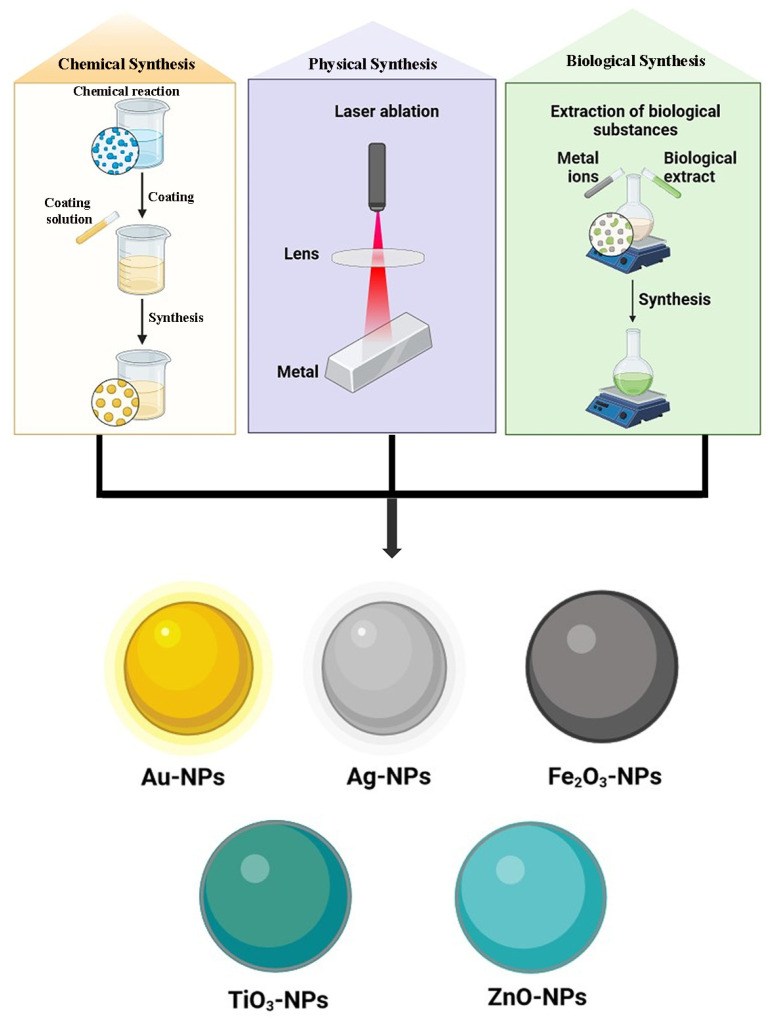
Various metallic nanoparticle systems of GA.

**Figure 11 pharmaceutics-16-01202-f011:**
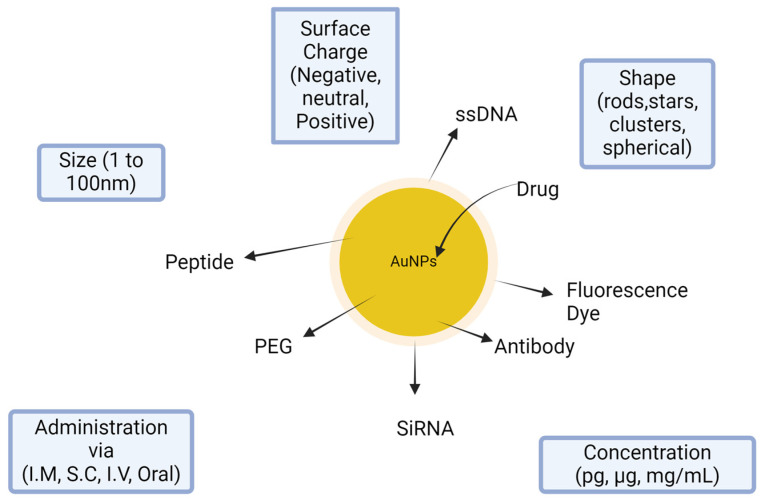
Salient features of gold nanoparticles in delivery system.

**Figure 12 pharmaceutics-16-01202-f012:**
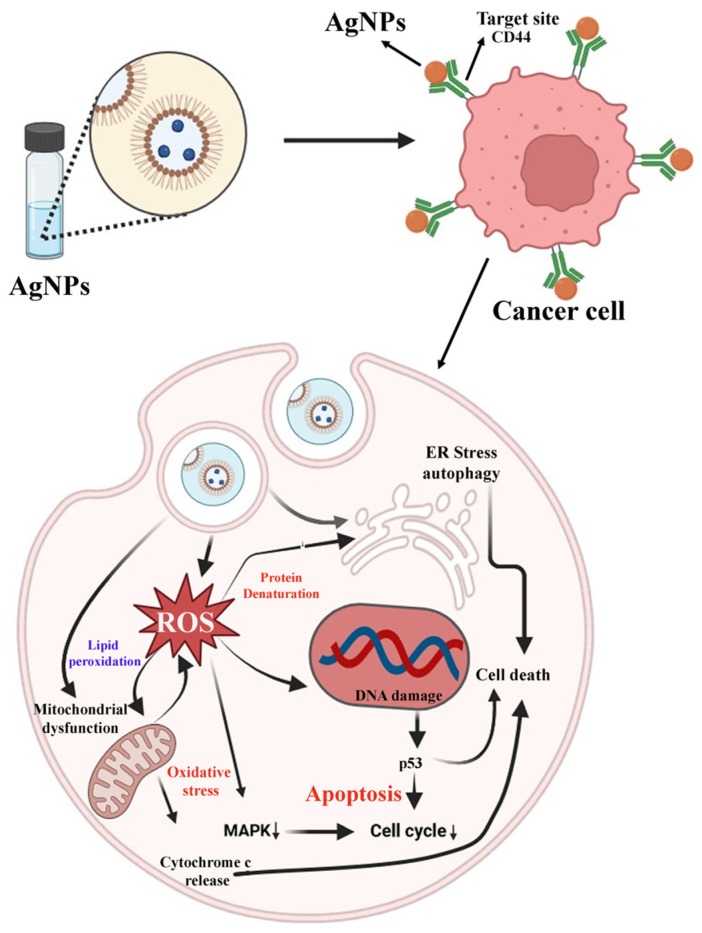
Anticancer potential of GA–AgNPs against osteosarcoma cells (SAOS-2) (↓ down regulation).

**Table 1 pharmaceutics-16-01202-t001:** Various novel drug delivery systems of gallic acid.

S. No.	Formulations	Physiochemical Characteristics	Method	Outcome	References
1	Liposomes	Av. Size: 181.5 nm, %EE: 98.3%, zeta potential: −53 mV, PDI < 0.3	Lipid film hydration method	The produced nanoscale lipid vesicles, with an average size of 181.5 nm, the highest percentage of EE (98.3%), and the highest rate of release (38.3%), were shown to be stable at 0.1 mg/mL of GA.	[88]
		Av. Size: 126 nm, %EE: 29%, zeta potential: 0.4 mV, PDI = 0.18	lipid film hydration, reverse-phase evaporation, ethanol injection, and ethanol permeabilization.	The GA-loaded Transferrin-functionalized liposome displayed beneficial properties such as neutral zeta potential, compact size, and reduced PDI.	[52]
		Av. Size: 153.2 ± 1.4 nm, %EE: 97.3 ± 2.5%, PDI: 0.252 ± 0.002	Thin-layer dispersion method	Compared with GA–LIP, LF–GA–LIP demonstrated superior antibacterial activity and better storage stability.	[89]
		Av. Size: 70 nm	-	The outcomes demonstrated that their combined antioxidant and antibacterial properties improved when resveratrol and gallic acid were co-loaded into vesicles.	[90]
		Particle size: 400 nm, %EE: 84.79%, zeta potential: −0.5 mV, PDI: 0.23	Thin film hydration	The nanogallic acid and co-loaded particles in breast and lung cancer cell lines displayed the best predicted IC_50_ values. The formula for nanoquercetin showed the least amount of cytotoxicity.	[91]
		Av. Size: 71.7 nm, zeta potential: −10.81 mV PDI: 0.286	Ethanol injection method	It was discovered that encapsulating (-)-epigallocatechin-3-gallate (EGCG) in nanoliposomes significantly improved its stability during in vitro digestion, increasing its bioavailability and maybe improving its health.	[92]
		Particle size: 400 nm, %EE: 84.79%, zeta potential: −0.5 mV, PDI = 0.23	Thin film hydration	The nanogallic acid and co-loaded particles in breast and lung cancer cell lines displayed the best predicted IC_50_ values. The formula for nanoquercetin showed the least amount of cytotoxicity.	[91]
		Av. Size: 71.7 nm, zeta potential: −10.81 mV, PDI: 0.286	Ethanol injection method	It was discovered that encapsulating EGCG in nanoliposomes significantly improved its stability during in vitro digestion, increasing its bioavailability and maybe improving its health.	[92]
2	Solid lipid nanoparticles	Particle Size: 224.40–3596.30 nm, %EE: 93.75% PDI: 0.85	Double emulsion method, melt dispersion technique.	For fortification, four distinct SLN–gallic acid concentrations were tested: 0, 2.5, 5, 7.5, etc.	[93]
		Average size: 120 ± 8.8 nm, PDI: 0.12 ± 0.08, zeta potential: −38 ± 10.2 mV	Solvent injection technique	The creation and characterization of a hydrogel loaded with n-propyl gallate-encased solid lipid nanoparticles for intranasal delivery.	[94]
3	Nanostructured lipid carriers	Particle Size: 169.46 nm, %EE: 90.06%, zeta potential: −36.39 mV PDI: 0.212	Solvent injection technique	Synthesis of NLCs to encapsulate coenzyme Q10 and extract from *Myrica esculenta* leaves, showcasing the formulation’s optimized tiny particle size and high entrapment effectiveness.	[95]
4	Nanoemulsions	Particle size: 162.1 nm, %EE: 97.3% ± 2.7, zeta potential: −46.5 ± 1.3 mV, PDI: 0.11	Spontaneous emulsification method	The created formulations displayed non-Newtonian pseudoplastic behavior, a high zeta potential, appropriate PG concentration, and nanometric droplet sizes.	[46]
5	Transferosomes	Particle size: 100 nm, %EE: 1%, PDI: 0.1	Thin film hydration method	The study demonstrated that with effective transdermal delivery of gallic acid, the optimized transfersomal formulation might be employed to prevent and treat cutaneous diseases.	[62]
6	Niosomes	Particle size: 102.33–143.77 nm, %EE: 4.48 ± 2.10%, zeta potential: 79.87 ± 0.72 mV, PDI: 0.28 ± 0.01	Film hydration technique	The most excellent anti-skin aging efficacy was shown by the gallic acid-loaded cationic CTAB niosome, which also exhibited antioxidant activity, inhibition of matrix metalloproteinase-2, melanin suppression, and tyrosinase and tyrosinase-related protein-2 inhibition.	[96]
		Particle size: 235.07 ± 7.43 nm, %EE: −46.64 ± 0.78%, zeta potential: −46.64 ± 0.78 mV, PDI: 2.50 ± 0.78	Sonication	The study developed elastic and non-elastic niosomes loaded with gallic acid or the semi-purified fraction isolated from *Terminalia chebula* galls.	[65]
		Av. Size: 80.2 ± 5.7 nm, %EE: 75 ± 3%, zeta potential: −44.78 ± 2.3 mV, PDI: 0.26 ± 0.01	Thin film hydration method	The study found that the GAN formulation, or F2 formulation, had the potential to be a successful vehicle for GA that exhibited anti-tyrosinase, anti-microbial, and anti-melanoma properties.	[97]
7	Dendrimers	Particle size: 349.9 ± 1.08 nm, %EE: 48.4, zeta potential: −29.2 ± 0.7 mV, PDI: 0.708 ± 0.023	-	The paper’s primary goal is to investigate the possibility of GA and GA–dendrimer formulations in combating chemo-resistant neuroblastoma cells.	[81]
		Av. Size: 387 ± 11 nm, zeta potential: −25 ± 0.34 mV	Esterification	The development of potent antioxidant-exhibiting gallic acid-shelled dendrimer nanoparticles that are biodegradable and biocompatible is discussed in the study.	[98]
8	Nanocapsules	Particle size: 30.35 ± 2.34 nm, %EE: 63.95 ± 2.98%, zeta potential: −13.90 ± 0.35 mV to 11.80 ± 0.22 mV PDI: 0.22 ± 0.02 and 0.50 ± 0.01,	Response surface methodology	Compared with GA alone, GA–RLMNC showed more significant apoptosis, downregulated expressions of profibrogenic indicators, successful internalization into an HCS, and higher proliferative activity.	[77]
		%EE: 79%	-	The created system demonstrated pH-dependent release behavior, making it suitable for treating cancer cells due to their lower pH medium than normal cells.	[99]
		Particle size: 259 to 268 nm, %EE: 16%, zeta potential: −11.43 to −15.98 mV, PDI: 0.361 to 0.390	Ultrasonication	The nanoencapsulation significantly impacted the loaded nanocapsule’s colloidal characteristics.	[100]
9	Micelles	Dipole moment: 1.77, Absorbance Maxima: 260 nm, Emission maxima: 339 nm, FQY: 0.010	Spectroscopic analysis	According to the study, there is a pH-dependent shift in the absorbance maxima of GA, with a change between 260 nm at pH 7 and 272 nm at pH 2.8.	[101]
		Av. Size: 171–203 nm, PDI = 0.48 ± 0.04	Solvent casting method	This study concludes that NAR1 and NAR2 PMs have lower particle sizes than GA1 and GA2 PMs and have the desired physicochemical features of novel drug delivery vehicles.	[76]
10	Nanogels	Particle size: 573.3 ± 207.2 nm, %EE: 70.94%,	Atom transfer radical polymerization	It was discovered that the GA-loaded nanogels were biocompatible with human cervical cancer cells.	[102]
11	Ethosomes	-	Mechanical dispersion, ethanol injection method, film hydration method	The non-ionic ethosomal technique was significantly more successful at delivering and promoting the adsorption of black tea extracts to the hair surface than hydroalcoholic extracts.	[103]

Av. Size: average size, PDI: poly dispersity index, FQY: fluorescence quantum yield, %EE: %entrapment efficiency.

**Table 3 pharmaceutics-16-01202-t003:** Marketed formulations of gallic acid.

Brand Name	Company Name	Composition	Dosage Form	Application
Fix Derma	Fix Derma India Private Limited, Gurugam, India	5% Niacinamide + mandelic acid 6.5% + gallic acid 0.5%	Serum	Skin ageing
Gyno-Go	Swiss Beauty, Delhi, India	Wild oregano essence + caffeine extract+ ginger infusion + retinol + ginkgo biloba extract + green tea infusion	Effervescent Tablet	Gynaecomastia
Aminu	Aminu Wellness Private Limited, Mumbai, India	Diglucosyl gallic acid + mulberry + pine mushroom extracts + Niacinamide + acetyl glucosamine	Liquid drops	Hyperpigmentation

**Table 4 pharmaceutics-16-01202-t004:** Patent on gallic acid and its derivatives.

S. No.	Patent Number	Title	Patent Year
1	US3560569A	Preparation of gallic acid	1971
2	CA2215251C	Industrial preparation of high-purity gallic acid	2003
3	US-2022356141-A1	Method for producing a composition containing gallic acid	2022
4	EP1437119B1	External preparation containing gallic acid derivatives	2007
5	US4968438A	Gallic acid as an oxygen scavenger	1990
6	CN108929220A	A kind of preparation method for gallic acid	2021
7	EP3562830B1	Hydroxybenzoic acid derivatives, methods, and uses thereof	2023
8	EP2510797A1	Method for producing purified tea extract	2014
9	US6472190B1	Biocatalytic synthesis of galloid organics	2002

**Table 5 pharmaceutics-16-01202-t005:** Clinical trials on gallic acid.

S. No.	Title	Phase	NCT Number	Status	Interventions	Actual Enrollment	Intervention Model	Location
1	Role of Ajwa Derived Polyphenols in Dyslipidaemias	-	NCT03805139	Unknown	Dietary supplement: ajwadate (*Phoenix dactylifera*)	60 participants	-	Karachi, Sindh, Pakistan

## Data Availability

Data can be made available at the request of correspondence authors.

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
