# Peer review of "Recent Advancements in Gallic Acid-Based Drug Delivery: Applications, Clinical Trials, and Future Directions"

_pharmaceutics, 2024, doi:10.3390/pharmaceutics16091202_

Round 1

Reviewer 1 Report

Comments and Suggestions for Authors

GA is a well-known herbal bioactive compound found in many foods. It has various pharmacological activities, but its low permeability and bioavailability limit its efficacy. To address this issue, novel drug delivery approaches have been developed to improve the stability of GA. Nano-carrier systems have been developed for GA. The article is of good quality and clear. I recommend this paper to be published in the journal. Here are some minor suggestions:

1.     Make sure all abbreviations are written out in full the first time used (such as GSH in page 4; NDDS [nano drug delivery systems (NDDS)]…). This is particularly important in the abstract (such as BCS), but work through the entire ms carefully from this perspective.

2.     There is a lack of recent literature citations. Authors also should cite some literatures published in 2024. “Gallic acid (GA) is a polyphenolic herbal molecule, has been reported for several health benefits. GA and its derivatives are present in medicinal plants, have been famous for their several pharmacological activities. (DOI: 10.1016/j.biomaterials.2024.122724)”; “The bioavailability of metoprolol through the oral cavity was dramatically increased by gallic and ellagic acid, which inhibited the CYP2D6-mediated process of metabolism in the rat liver (?)”; “In this context, NDDS has pivotal role in solving the issue of solubility, permeability, and therapeutic efficacy. (DOI: 10.1016/j.phrs.2024.107150)”; “Oxidative stress can result from an imbalance between the body's ability to remove and purify reactive oxygen compounds (ROS) and the quantity of ROS generated. Reactive oxygen intermediates include anion superoxide (O*-), peroxide of hydrogen (H2O2), and radicals of hydroxyl (HO*).(DOI: 10.3390/biom14010128)” .

3.     “Despite various materials being explored as carriers for co-delivery, challenges regarding drug loading capacity, stability, and biocompatibility persist. Therefore, GA-based carrier-free NPs with exceptionally high chemotherapeutic drug content have emerged as a new trend in drug delivery research. Carrier-free nanoparticulate drug delivery systems offer distinct advantages, such as long circulation time, high tumor targeting efficiency, and controllable drug release. (doi:10.1016/j.jcis.2024.08.100)” Please add relevant content (Carrier-free nanoparticulate drug delivery systems) to the conclusion to highlight future trends.

Comments on the Quality of English Language

Minor editing of English language required.

Author Response

GA is a well-known herbal bioactive compound found in many foods. It has various pharmacological activities, but its low permeability and bioavailability limit its efficacy. To address this issue, novel drug delivery approaches have been developed to improve the stability of GA. Nano-carrier systems have been developed for GA. The article is of good quality and clear. I recommend this paper to be published in the journal. Here are some minor suggestions:

Ans: Thanks for your observation and valuable suggestion. Now this time manuscript has been revised properly as suggested by the editor and reviewers.

  1. Make sure all abbreviations are written out in full the first time used (such as GSH in page 4; NDDS [nano drug delivery systems (NDDS)]…). This is particularly important in the abstract (such as BCS), but work through the entire MS carefully from this perspective.

Ans: Thank you very much for observation and suggestion. Manuscript has been thoroughly checked for abbreviations and their full form have been provided at first appearance in the text, including abstract (page 1).

  1. There is a lack of recent literature citations.Authors also should cite some literatures published in 2024. “Gallic acid (GA) is a polyphenolic herbal molecule, has been reported for several health benefits. GA and its derivatives are present in medicinal plants, have been famous for their several pharmacological activities. (DOI: 10.1016/j.biomaterials.2024.122724)”; “The bioavailability of metoprolol through the oral cavity was dramatically increased by gallic and ellagic acid, which inhibited the CYP2D6-mediated process of metabolism in the rat liver (?)”; “In this context, NDDS has pivotal role in solving the issue of solubility, permeability, and therapeutic efficacy. (DOI: 10.1016/j.phrs.2024.107150)”; “Oxidative stress can result from an imbalance between the body's ability to remove and purify reactive oxygen compounds (ROS) and the quantity of ROS generated. Reactive oxygen intermediates include anion superoxide (O*-), peroxide of hydrogen (H2O2), and radicals of hydroxyl (HO*).(DOI: 10.3390/biom14010128)” .

Ans: Thanks a lot for your observation and suggestion. References have been ctied in the manuscript as suggested by the reviewer. Detail is as follows:

  • DOI: 10.1016/j.biomaterials.2024.122724 [1], is cited in page 2.
  • DOI: 10.1016/j.phrs.2024.107150 [11], is cited in page 3.
  • DOI: 10.3390/biom14010128 [23], is cited in page 4.

  1. “Despite various materials being explored as carriers for co-delivery, challenges regarding drug loading capacity, stability, and biocompatibility persist. Therefore, GA-based carrier-free NPs with exceptionally high chemotherapeutic drug content have emerged as a new trend in drug delivery research. Carrier-free nanoparticulate drug delivery systems offer distinct advantages, such as long circulation time, high tumor targeting efficiency, and controllable drug release. (doi:10.1016/j.jcis.2024.08.100)” Please add relevant content (Carrier-free nanoparticulate drug delivery systems) to the conclusion to highlight future trends.

Ans: Thanks for suggestion. Necessary information has been provided in the Section 7 (page 31).

Comments on the Quality of English Language

Ans: Thank you so much for your suggestion. Manuscript has been checked thoroughly for English language and edited properly wherever needed.

Minor editing of English language required.

Ans: Thank you very much for your valuable observation. Now we have revised the manuscript for typographical error, and improved the English language throughout the manuscript.

Reviewer 2 Report

Comments and Suggestions for Authors

The work presented in this article provides sufficient data of interest to the journal. However, several changes are needed to enhance its quality and make it an excellent piece of work:

  1. It would be beneficial for the audience to provide the full name of BCS class III.
  2. In Section 4 (Different Nanocarrier Systems of GA), it would be better to include a table that addresses all previously prepared Nanocarrier Systems of GA. The table should include the name of each preparation, its uses, references, and whether it has been approved by the FDA.
  3. Figures 5 and 6 are not suitable as they have been previously published. It would be better to create graphical images illustrating the mechanism of gallic acid in wound healing and its anticancer effects on the lungs.
  4. Figure 7 should be replaced with a graphical image that describes the role of gallic acid in treating nephrotoxicity.
  5. Figure 8 should also be replaced with a graphical image showing the cytotoxic effects of gallic acid.
  6. For the potential effect, role, and mechanism of both GA and Ag in the treatment of osteosarcoma cells, it would be better to provide a graphical image instead of Figure 10.

Author Response

The work presented in this article provides sufficient data of interest to the journal. However, several changes are needed to enhance its quality and make it an excellent piece of work:

  1. It would be beneficial for the audience to provide the full name of BCS class III.

Ans: Thank you so much for your observation. Now full form of BCS (Biopharmaceutics classification system) has been provided in the abstract (page 1) and manuscript (page 2).

  1. In Section 4 (Different Nanocarrier Systems of GA), it would be better to include a table that addresses all previously prepared Nanocarrier Systems of GA. The table should include the name of each preparation, its uses, references, and whether it has been approved by the FDA.

Ans: Thanks for your observation and suggestion. In section 4, already a Table 1 has been provided. As per your suggestion necessary information has been included in Table 1 (refer pages 17-20).

  1. Figures 5 and 6 are not suitable as they have been previously published. It would be better to create graphical images illustrating the mechanism of gallic acid in wound healing and its anticancer effects on the lungs.

Ans: Thank you very much for observation and suggestion. Figure 5 and 6 has been modified as graphical image (revised Figure 6) and provided in the manuscript as suggested by the reviewer.

  1. Figure 7 should be replaced with a graphical image that describes the role of gallic acid in treating nephrotoxicity.

Ans: Thank you very much for observation. Figure 7 has been designed as graphical image and provided in the manuscript as suggested.

  1. Figure 8 should also be replaced with a graphical image showing the cytotoxic effects of gallic acid.

Ans: Thanks for observation and suggestion. Figure 8 has been designed as graphical image and given in the manuscript.

  1. For the potential effect, role, and mechanism of both GA and Ag in the treatment of osteosarcoma cells, it would be better to provide a graphical image instead of Figure 10.

Ans: Thanks a lot for observation. Figure 10 has been designed as a graphical image and provided in the manuscript as suggested.

Reviewer 3 Report

Comments and Suggestions for Authors

1. The chemical formula for gallic acid is 3,4,5-trihydroxybenzoic acid. The chemical structure in Figure 1 is missing the hydroxyl group at the 4-position. Please correct this by adding the OH group at the 4-position.

Nanocarrier Preparation and Nanocarrier Systems of GA

2. Discuss the limitations and challenges of the nanocarrier systems' preparation, toxicity, stability, and bioavailability. Additionally, comparing the merits and drawbacks of different NP systems would significantly enhance the support for this article.

3. If possible, provide a general diagram (or formula) for each type of NPs preparation process, as this would likely help the reader better understand the content.

4. The clinical trial data is limited, especially in gallic acid nanomedicines. I would suggest that the authors present more clinical developments, such as ongoing trials or potential applications, and prospects, such as future research directions or market potential.

General suggestions

5. Enhance the quality of the figures to improve the clarity of the information.

Author Response

  1. The chemical formula for gallic acid is 3,4,5-trihydroxybenzoic acid. The chemical structure in Figure 1 is missing the hydroxyl group at the 4-position. Please correct this by adding the OH group at the 4-position.

Ans: Thank you very much for your observation. Now Figure 1 has been revised as suggested by the reviewers. Gallic acid structure and -OH in 4th position has been corrected (page no. 2).

Nanocarrier Preparation and Nanocarrier Systems of GA

  1. Discuss the limitations and challenges of the nanocarrier systems' preparation, toxicity, stability, and bioavailability. Additionally, comparing the merits and drawbacks of different NP systems would significantly enhance the support for this article.

Ans: Thank you very much for observation and suggestion. We have revised the manuscript and included the relevant text inside the manuscript (section 7, last para) as suggested by the reviewers.

  1. If possible, provide a general diagram (or formula) for each type of NPs preparation process, as this would likely help the reader better understand the content.

Ans: Thank you very much for suggestion. The necessary information and general diagrma for developing nanoparticles have been provided in Figure 5 (page 8-9) and Figure 10 (page 23).

  1. The clinical trial data is limited, especially in gallic acid nanomedicines. I would suggest that the authors present more clinical developments, such as ongoing trials or potential applications, and prospects, such as future research directions or market potential.

Ans: Thank you very much for suggestion. Again, we have accessed the NIH, clinical trials site (https://clinicaltrials.gov/) dated 06-09-2024, for getting information about gallic acid, but only single result has shown which has already explained in Section 6, Table 5, page 32.

General suggestions

  1. Enhance the quality of the figures to improve the clarity of the information.

Ans: Thank you very much for suggestion. Now in the revised version form of manuscript, all the figures quality has been improved individually.

Round 2

Reviewer 2 Report

Comments and Suggestions for Authors

thanks to authors, all comment's have been changed and addressed

I dont have any further comment's